# Genetically prolonged beige fat in male mice confers long-lasting metabolic health

Ruifan Wu [1,2,11], Jooman Park[1,11], Yanyu Qian[1], Zuoxiao Shi[1,3], Ruoci Hu[1,3], Yexian Yuan[1,2], Shaolei Xiong[4], Zilai Wang[4], Gege Yan[5], Sang-Ging Ong [5,6], Qing Song[7], Zhenyuan Song [7], Abeer M. Mahmoud[8], Pingwen Xu [8], Congcong He [9], Robert W. Arpke [10], Michael Kyba [10], Gang Shu [2], Qingyan Jiang [2] & Yuwei Jiang [1,3,8] ✉

A potential therapeutic target to curb obesity and diabetes is thermogenic beige adipocytes. However, beige adipocytes quickly transition into white adipocytes upon removing stimuli. Here, we define the critical role of *cyclin dependent kinase inhibitor 2A (Cdkn2a)* as a molecular pedal for the beige-to-white transition. Beige adipocytes lacking *Cdkn2a* exhibit prolonged lifespan, and male mice confer long-term metabolic protection from diet-induced obesity, along with enhanced energy expenditure and improved glucose tolerance. Mechanistically, *Cdkn2a* promotes the expression and activity of beclin 1 (BECN1) by directly binding to its mRNA and its negative regulator BCL2 like 1 (BCL2L1), activating autophagy and accelerating the beige-to-white transition. Reactivating autophagy by pharmacological or genetic methods abolishes beige adipocyte maintenance induced by *Cdkn2a* ablation. Furthermore, hyperactive BECN1 alone accelerates the beige-to-white transition in mice and human. Notably, both *Cdkn2a* and *Becn1* exhibit striking positive correlations with adiposity. Hence, blocking *Cdkn2a*-mediated BECN1 activity holds therapeutic potential to sustain beige adipocytes in treating obesity and related metabolic diseases.

Obesity and its associated metabolic diseases, resulting from chronic energy imbalance, have rapidly prevailed over the past decades worldwide and have become a global public health issue[1]. In mammals, adipocytes are an important regulator of energy homeostasis by storing and releasing energy. White adipocytes store energy as one large lipid droplet, while brown adipocytes with multilocular lipid droplets and abundant mitochondria can dissipate energy to produce heat mediated by uncoupling protein (UCP1)[2]. Notably, beige adipocytes are a subset of "brown-like" cells in white adipose tissue (WAT) that also express UCP1 and contain plentiful mitochondria, thereby possessing thermogenic capability[3]. Although brown adipocytes are limited to infants, beige adipocytes can be recruited throughout WAT

[1]Department of Physiology and Biophysics, College of Medicine, University of Illinois at Chicago, Chicago, IL 60612, USA. [2]Guangdong Laboratory of Lingnan Modern Agriculture, Guangdong Province Key Laboratory of Animal Nutritional Regulation and National Engineering Research Center for Breeding Swine Industry, College of Animal Science, South China Agricultural University, Guangzhou 510642, China. [3]Department of Pharmaceutical Sciences, University of Illinois at Chicago, Chicago, IL 60612, USA. [4]Department of Microbiology and Immunology, University of Illinois at Chicago, Chicago, IL 60612, USA. [5]Department of Pharmacology and Regenerative Medicine, College of Medicine, University of Illinois at Chicago, Chicago, IL 60612, USA. [6]Division of Cardiology, Department of Medicine, University of Illinois at Chicago, Chicago, IL 60612, USA. [7]Department of Kinesiology and Nutrition, University of Illinois at Chicago, Chicago, IL 60612, USA. [8]Division of Endocrinology, Department of Medicine, University of Illinois at Chicago, Chicago, IL 60612, USA. [9]Department of Cell and Developmental Biology, Feinberg School of Medicine, Northwestern University, Chicago, IL 60611, USA. [10]Lillehei Heart Institute, University of Minnesota, Minneapolis, MN 55455, USA. [11]These authors contributed equally: Ruifan Wu, Jooman Park. ✉e-mail: yuweij@uic.edu

by various mechanisms in adult humans, which confers metabolic benefits, including reduced blood glucose and increased energy expenditure[4–6]. Thus, beige adipocytes have emerged as a promising cellular target as a metabolic sink for glucose and free fatty acids to combat obesity and diabetes.

In both rodents and humans, the biogenesis of beige adipocytes is highly inducible by a list of physiological and pharmacological stimuli, such as chronic cold exposure or β3 adrenergic receptor agonists[7]. However, beige adipocytes are short-lived; after removing external stimuli, UCP1+ multilocular beige adipocytes convert into dormant unilocular "white" adipocytes, expressing several white adipocyte-enriched genes[8–11]. Thus, constant stimuli are needed to maintain the activity of beige adipocytes, which has become one of the major clinical hurdles for using beige adipocytes as a sustainable therapy for chronic metabolic diseases. Compared to beige adipocyte triggering, relatively little research has been conducted to investigate the fates of beige adipocytes after discontinuing beiging stimuli. Therefore, it is critical to elucidate the precise mechanism(s) by which beiging is maintained and to examine whether maintaining beiging can improve metabolic fitness.

The genomic *Cdkn2a* locus encodes two crucial cell cycle regulators, p16[Ink4a] and p14[ARF], which regulate the p16[Ink4a]/RB and p14[ARF]/p53 pathways, respectively. The role of *Cdkn2a* as a tumor suppressor gene is well known[12–14]. Recently, *Cdkn2a* has been linked to obesity and type 2 diabetes in both rodent and human genome-wide association studies[15,16]. Studies have demonstrated that *Cdkn2a* expression is increased in WAT in obese mouse models and obese humans, inducing senescence and inflammation and aggravating systemic insulin resistance[17–19]. Whole-body *Cdkn2a* knockout mice gained less weight and fat under a high-fat diet, along with increased beige fat formation in WAT and energy expenditure[19]. Consistent with these studies, our recent study showed that the deletion of *Cdkn2a* in UCP1-positive cells within inguinal WAT (IGW) stimulated beige adipocyte proliferation[20]. These findings suggest that *Cdkn2a* plays a crucial role in obesity and diabetes. However, whether *Cdkn2a* regulates beige fat maintenance is unknown.

In this study, we investigated the function of *Cdkn2a* in beige fat maintenance by employing our defined *Cdkn2a* mouse models and in vitro beige maintenance assays. We found that loss of *Cdkn2a* in beige adipocytes results in prolonged beige adipocyte maintenance, enhanced energy expenditure, improved glucose tolerance, and increased resistance to high-fat, high-sucrose (HFHS) diet-induced obesity. Mechanistically, *Cdkn2a* deficiency promotes beige adipocyte maintenance by inhibiting BECN1-mediated autophagy. Importantly, reactivating autophagy in *Cdkn2a*-ablated beige adipocytes reversed beige adipocyte maintenance phenotypes. Consistent with a role in energy expenditure, both *Cdkn2a* and *Becn1* exhibit striking positive correlations with adiposity in mice and humans. Collectively, our findings unveil insights into the molecular mechanisms by which *Cdkn2a* regulates beige adipocyte maintenance via BECN1-mediated autophagy and provide a potential therapeutic strategy to lock into a stable beige identity and treat obesity and related metabolic diseases.

## Results

### *Cdkn2a* deficiency maintains the morphological and molecular characteristics of beige adipocytes after withdrawal of cold exposure

To investigate the function of *Cdkn2a* in beige fat maintenance in vivo, we crossed tamoxifen (TAM)-inducible *Ucp1*-Cre[ERT2] mice with a *Rosa*26R[RFP] indelible labeling reporter and *Cdkn2a* floxed mice (*Cdkn2a*[Ucp1] KO, Fig. 1a). Using this fate-mapping model, we first induced beige adipocyte biogenesis in IGW for 7 days by cold exposure (6 °C) to 2-month-old male mice. After cold exposure, we deleted *Cdkn2a* in all newly induced and preexisting UCP1-expressing beige adipocytes through TAM administration (1.5 mg/kg) for two

consecutive days. We housed the mice at room temperature (22 °C) for 60 days (rewarming period) to trace beige adipocyte maintenance (Fig. 1b). No significant difference was found in body weight or body composition between control and *Cdkn2a*[Ucp1] KO mice after the rewarming period following cold exposure (Fig. 1c and Supplementary Fig. 1a). As expected, at day 0, all RFP+ beige adipocytes within IGW expressed endogenous UCP1 following cold exposure, as assessed by RFP, UCP1, and PLIN immunofluorescence staining (Fig. 1d, e). Consistent with a previous study that showed that beige adipocytes transition back to their white adipose state[10], we found that at day 60, nearly 80% of beige adipocytes lost UCP1 expression, as quantified by RFP+, UCP1−, and PLIN+ immunofluorescence staining (Fig. 1d, e). In contrast, *Cdkn2a*[Ucp1] KO mice contained a higher percentage of UCP1-expressing RFP+ beige adipocytes in IGW compared to control mice at 60 days post withdrawal of cold stimulus and TAM administration (Fig. 1d, e). Consistent with UCP1 immunostaining, hematoxylin and eosin (H&E) staining showed that beige fat biogenesis, determined by multilocular lipids, was similar at day 0 (Fig. 1f), but *Cdkn2a*[Ucp1] KO mice contained more beige adipocytes at day 60 (Fig. 1g). Moreover, *Cdkn2a* deletion reduced adipocyte size in IGW at day 60 (Fig. 1g, h). At the molecular level, we first confirmed that p16[Ink4a] and p19[Arf], two transcript variants of *Cdkn2a*, were deleted in IGW beige fat by qPCR analysis (Fig. 1i). There was no significant difference in the basal expression levels of *Ucp1* and other thermogenic genes, such as *Ppargc1a*, *Pparg*, and *Cidea*, in IGW between control and *Cdkn2a*[Ucp1] KO mice at day 0 (Fig. 1i). In agreement with beige adipocyte maintenance, loss of *Cdkn2a* increased thermogenic gene expression in IGW at day 60 compared to the control group (Fig. 1j). Furthermore, we observed a similar pattern of protein abundance of UCP1: no difference at day 0 but significantly higher in IGW from *Cdkn2a*[Ucp1] KO mice after the rewarming period (Fig. 1k). Similar to the observations in male mice, female *Cdkn2a*[Ucp1] KO mice also exhibited increased beige adipocyte content (Supplementary Data Fig. 1b) and higher expression levels of thermogenic genes at day 60 compared to control mice (Supplementary Data Fig. 1c). This suggests that Cdkn2a knockout promotes beige adipocyte maintenance in both male and female mice. To investigate whether increased sympathetic neural signaling was responsible for the prolonged beige fat maintenance in *Cdkn2a*[Ucp1] KO mice, we performed western blot against tyrosine hydroxylase (TH), which is a marker of norepinephrine turnover. However, no significant difference in TH expression was observed in IGW between control and *Cdkn2a*[Ucp1] KO mice at day 60 (Fig. 1l).

We next evaluated whether deleting *Cdkn2a* in UCP1+ cells impacted the morphological and molecular characteristics of brown adipocytes and other organs. We observed no differences in interscapular brown adipose tissue (BAT) morphology between control and *Cdkn2a*[Ucp1] KO mice based on H&E analysis (Supplementary Fig. 1d). In contrast to beige adipocytes, brown adipocytes from both control and *Cdkn2a*[Ucp1] KO mice expressed constitutively high levels of UCP1 even at 60 days after the rewarming period following cold exposure (Supplementary Fig. 1e–g), which was consistent with a previous study[10]. Moreover, with qPCR analysis, we confirmed that *Cdkn2a* (p16[Ink4a] and p19[Arf]) was largely ablated in BAT as expected, while thermogenic gene expression in BAT did not change compared to the control and *Cdkn2a*[Ucp1] KO mice at both day 0 and day 60 (Supplementary Fig. 1h). Furthermore, morphology and adipocyte size of perigonadal WAT (PGW) showed no significant difference between control and *Cdkn2a*[Ucp1] KO mice (Supplementary Fig. 1i, j). In addition, no significant difference was observed in liver histology (Supplementary Fig. 1k) and other tissue weights (Supplementary Fig. 1l) between control and *Cdkn2a*[Ucp1] KO mice. Overall, these results indicate that *Cdkn2a* promotes beige adipocyte maintenance but does not affect brown adipocytes or other organs after the rewarming period following cold exposure.

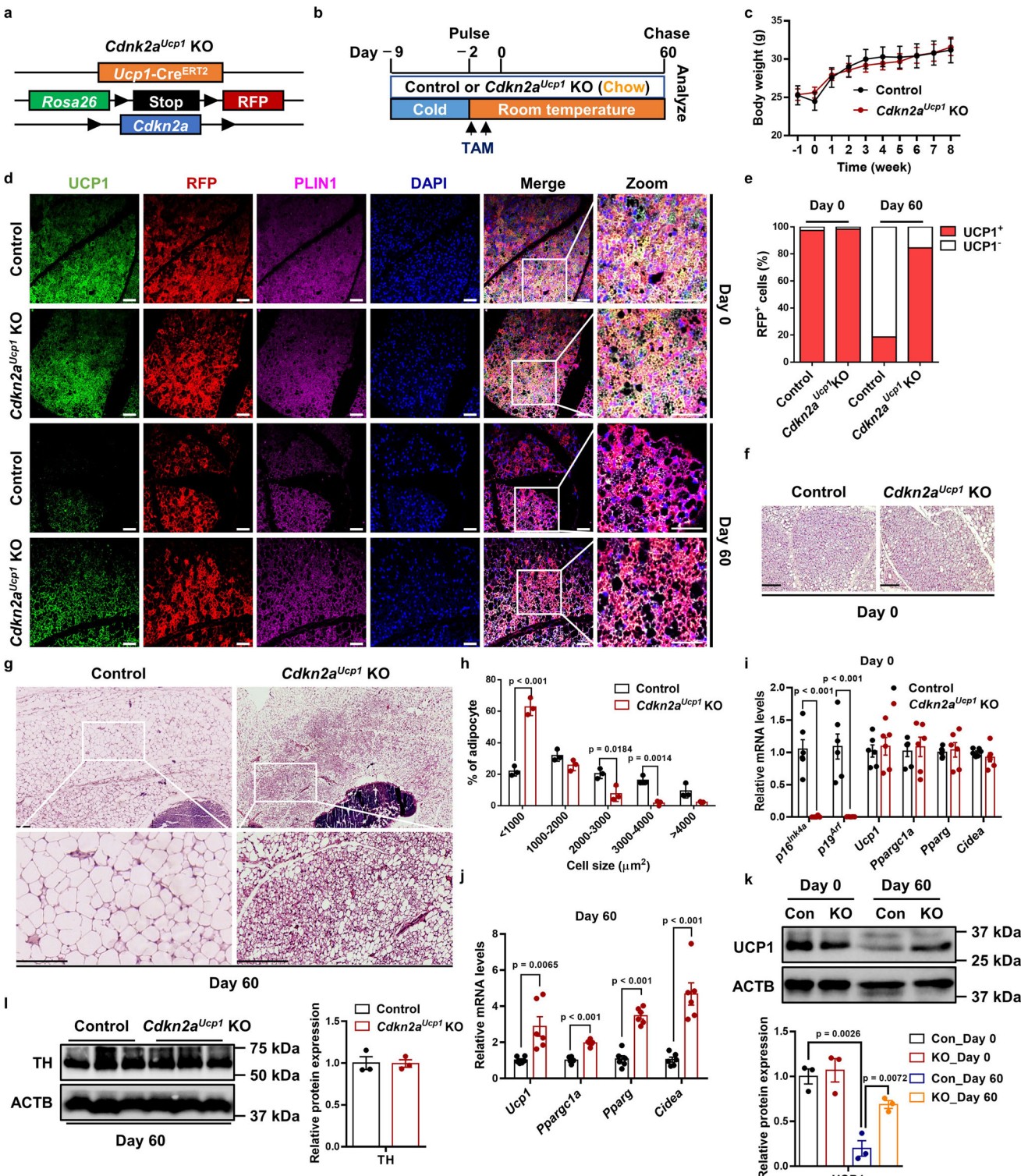

## Deleting *Cdkn2a* prevents HFHS-induced obesity and glucose intolerance by enhancing energy expenditure and thermogenesis

In both mice and humans, beige adipocytes can reduce body fat and glucose[21], presenting a key therapeutic promise. Our data thus far suggest that deleting *Cdkn2a* in beige adipocytes can extend their lifespan but without significant metabolic benefits under a standard chow diet. Chronic exposure to the HFHS diet, which mimics the human diet in many countries, can cause obesity, T2DM, and associated metabolic syndrome[22]. To further understand the functional

relevance of *Cdkn2a*-mediated beige adipocyte maintenance in obesity and its related diabetes, we challenged the control and *Cdkn2a^{Ucp1}* KO mice with a HFHS diet for 120 days during the rewarming period following cold exposure (Fig. 2a). Deleting *Cdkn2a* prevented HFHS-induced weight gain (Fig. 2b, c), while food intake showed no significant difference between control and *Cdkn2a^{Ucp1}* KO mice (Supplementary Fig. 2a). Compared to control mice, *Cdkn2a^{Ucp1}* KO mice exhibited markedly reduced body fat content and increased lean mass content (Fig. 2d). Furthermore, *Cdkn2a* deficiency mainly decreased the weight of WAT depots (IGW, PGW, retroperitoneal WAT (rpWAT))

**Fig. 1 | Ablation of *Cdkn2a* maintains beige adipocyte characteristics after the rewarming period following cold exposure. a** Scheme of *Cdkn2a^Ucp1* KO reporter mice used to conditionally knockout *Cdkn2a* in UCP1+ cells (*Ucp1*-Cre^ERT2; *Rosa*26R^RFP; *Cdkn2a*^fl/fl) and label with RFP. **b** Experimental procedure to track beige adipocytes in vivo. Two-month-old UCP1-RFP (control) and *Cdkn2a^Ucp1* KO male mice were housed under cold conditions for 7 consecutive days to induce beige fat formation. Mice were then administered tamoxifen (TAM) and fed a chow diet at room temperature for 60 days. **c** Body weight of male control (*n* = 6) or *Cdkn2a^Ucp1* KO mice (*n* = 8). **d** Immunofluorescence staining of UCP1 and RFP in IGW from control or *Cdkn2a^Ucp1* KO male mice at day 0 and day 60 post withdrawal of cold stimulus. Scale bar, 50 μM. **e** Quantification of the percentage of RFP+ cells that express endogenous UCP1. **f** Representative H&E staining images of IGW from control or *Cdkn2a^Ucp1* KO male mice at day 0 post withdrawal of cold stimulus. Scale

bar, 100 μM. **g** Tile scan of IGW from control or *Cdkn2a^Ucp1* KO male mice at day 60 post withdrawal of cold stimulus. Scale bar, 100 μM. **h** Quantification of adipocyte sizes in IGW from control or *Cdkn2a^Ucp1* KO male mice at day 60 post withdrawal of cold stimulus (*n* = 3). **i** qPCR analysis of the mRNA expression of *p16^Ink4a*, *p19^Arf* and thermogenic genes in IGW from control or *Cdkn2a^Ucp1* KO male mice at day 0 post withdrawal of cold stimulus (*n* = 6). **j** qPCR analysis of the mRNA expression of *p16^Ink4a*, *p19^Arf* and thermogenic genes in IGW from control or *Cdkn2a^Ucp1* KO male mice at day 60 post withdrawal of cold stimulus (*n* = 6). **k** Western blot analysis of UCP1 in IGW from control or *Cdkn2a^Ucp1* KO male mice at day 0 and 60 post withdrawal of cold stimulus (*n* = 3). **l** Western blot analysis of TH in IGW from control or *Cdkn2a^Ucp1* KO male mice at day 60 post withdrawal of cold stimulus (*n* = 3). *p* values were determined by two-tailed Student's *t* test. Data are expressed as means ± SEM.

but not the weight of other tissues, including BAT, muscle, kidney, spleen, pancreas, heart, and liver (Supplementary Fig. 2b). *Cdkn2a* knockout decreased adipocyte size in IGW (Fig. 2e, f) and PGW (Supplementary Fig. 2c, d). Furthermore, we found that *Cdkn2a* knockout ameliorated HFHS-induced whitening of BAT (Supplementary Fig. 2e) and fatty liver (Supplementary Fig. 2e). Compared to control mice, *Cdkn2a^Ucp1* KO mice exhibited significantly decreased blood glucose levels (Fig. 2g) and serum insulin concentrations (Fig. 2h) under HFHS conditions. Importantly, loss of *Cdkn2a* improved glucose metabolism and insulin sensitivity compared to control mice, as assessed by the glucose tolerance test (Fig. 2i, j) and insulin tolerance test (Fig. 2k, l). Taken together, these results indicate that prolonged maintenance of beige adipocytes by deleting *Cdkn2a* prevents HFHS-induced body weight and fat mass gain and its associated impaired glucose metabolism.

To further investigate the metabolic significance of retaining beige adipocytes in vivo, we next measured gas exchange and whole-body energy expenditure of control and *Cdkn2a^Ucp1* KO mice under HFHS in metabolic cages. We found that *Cdkn2a^Ucp1* KO mice showed increases in oxygen consumption (Supplementary Fig. 2f), carbon dioxide generation (Supplementary Fig. 2g), and energy expenditure (Fig. 2m) compared to control mice during both light and dark cycles. No significant difference was found in locomotor activity between control and *Cdkn2a^Ucp1* KO mice (Supplementary Fig. 2h). Consistent with the increased metabolic rate, *Cdkn2a^Ucp1* KO mice also exhibited a higher core body temperature at room temperature (Fig. 2n). As expected, the H&E staining results showed that *Cdkn2a* deletion preserved beige fat identity in IGW (Fig. 2e). Furthermore, *Cdkn2a^Ucp1* KO mice contained a higher percentage of UCP1-expressing RFP+ cells in IGW than control mice (Fig. 2o, p). Consistently, loss of *Cdkn2a* maintained higher levels of thermogenic gene expression and UCP1 protein expression and decreased mRNA levels of white adipocyte marker genes, such as *Adipoq*, *Fabp4*, and *Lep*, in IGW (Fig. 2q, r), indicating that *Cdkn2a* knockout prevented beige-to-white adipocyte transition. In contrast, there was no significant difference in the expression of UCP1 and other thermogenic genes in BAT between control and *Cdkn2a^Ucp1* KO mice (Supplementary Fig. 2i–l). These data indicate that prolonged maintenance of beige adipocytes by *Cdkn2a* deletion preserves energy expenditure and promotes thermogenesis under HFHS challenges.

### *Cdkn2a* negatively regulates beige adipocyte maintenance in vitro

To delineate the underlying cellular and molecular mechanisms, we established a cellular system to study beige adipocyte maintenance in vitro. We isolated stromal vascular fraction (SVF) cells from the IGW of TAM-uninduced control and *Cdkn2a^Ucp1* KO mice and cultured them to produce beige adipocytes in vitro. After *Cdkn2a* deletion and RFP labeling by 4-hydroxy-tamoxifen (4-OHT) treatment, we withdrew external stimuli by replacing the differentiation medium with DMEM containing 5% bovine serum albumin (BSA) at the initial stage (day 0).

Then we determined how many RFP+ cells were still beige adipocytes at the final stage (day 4) (Fig. 3a). We found that the mRNA levels of *p16^Ink4a* and *p19^Arf* were inversely associated with UCP1 levels among brown, beige, and white adipocytes (Supplementary Fig. 3a). Furthermore, the mRNA levels of *p16^Ink4a* and *p19^Arf* gradually increased after withdrawing external stimuli (Supplementary Fig. 3b). Together, these data suggest a negative regulatory role of *Cdkn2a* in beige adipocyte maintenance.

At the initial stage, both control and *Cdkn2a^Ucp1* KO cells exhibited similar amounts of multilocular lipid droplet formation and UCP1 expression (Fig. 3b–d). At the final stage, the number and size of lipid droplets and UCP1 expression were significantly decreased in control cells (Fig. 3b–d), recapitulating our in vivo system in which RFP+ beige adipocytes lose UCP1 expression. In contrast to control cells, *Cdkn2a^Ucp1* KO beige adipocytes contained more lipid droplets (Fig. 3b) and a higher percentage of UCP1-expressing RFP+ cells (Fig. 3c, d) at the final stage. Consistently, no difference was found in the basal expression levels of thermogenic genes between control and *Cdkn2a^Ucp1* KO beige adipocytes at the initial stage (Supplementary Fig. 3c). In contrast, at the final stage, *Cdkn2a^Ucp1* KO cells expressed higher levels of thermogenic genes, such as *Ucp1*, *Ppargc1a*, *Pparg*, *Prdm16*, and *Cidea*, than control cells (Fig. 3e). Furthermore, the UCP1 protein level was higher in *Cdkn2a^Ucp1* KO cells after withdrawing external stimuli than in control cells (Fig. 3f). Consistent with the higher amount of UCP1+ beige adipocytes in *Cdkn2a^Ucp1* KO, we found that *Cdkn2a* deficiency promoted basal mitochondrial respiration and maximal mitochondrial respiratory capacity of beige adipocytes at the final stage but not at the initial stage, as assessed by using a seahorse extracellular flux analyzer (Fig. 3g).

Next, we investigated the role of *Cdkn2a* in regulating brown adipocyte maintenance in vitro by using the same cellular system (Supplementary Fig. 4a). In contrast to beige maintenance, the expression levels of *p16^Ink4a* and *p19^Arf* were dramatically decreased during brown adipocyte maintenance (Supplementary Fig. 4b). Interestingly, the size and number of lipid droplets were decreased in both control and *Cdkn2a^Ucp1* KO brown adipocytes at the final stage (Supplementary Fig. 4c). No significant change was found between control and *Cdkn2a^Ucp1* KO brown adipocytes at either the initial or the final stage, based on morphology and UCP1 immunostaining (Supplementary Fig. 4c–e). Additionally, thermogenic gene expression was similar between the control and *Cdkn2a^Ucp1* KO cells during brown fat maintenance (Supplementary Fig. 4f, g). Collectively, these results indicate that loss of *Cdkn2a* facilitates the maintenance of beige adipocytes, but not brown adipocytes, in vitro.

### *Cdkn2a* regulates beige adipocyte maintenance in a cell cycle-independent manner

Since *Cdkn2a* is a well-known cell cycle inhibitor gene, we examined whether *Cdkn2a* regulated beige adipocyte maintenance in a cell cycle-dependent manner. Our previous study showed that deleting *Cdkn2a* in UCP1+ cells in IGW promotes beige adipocyte proliferation[20]. To

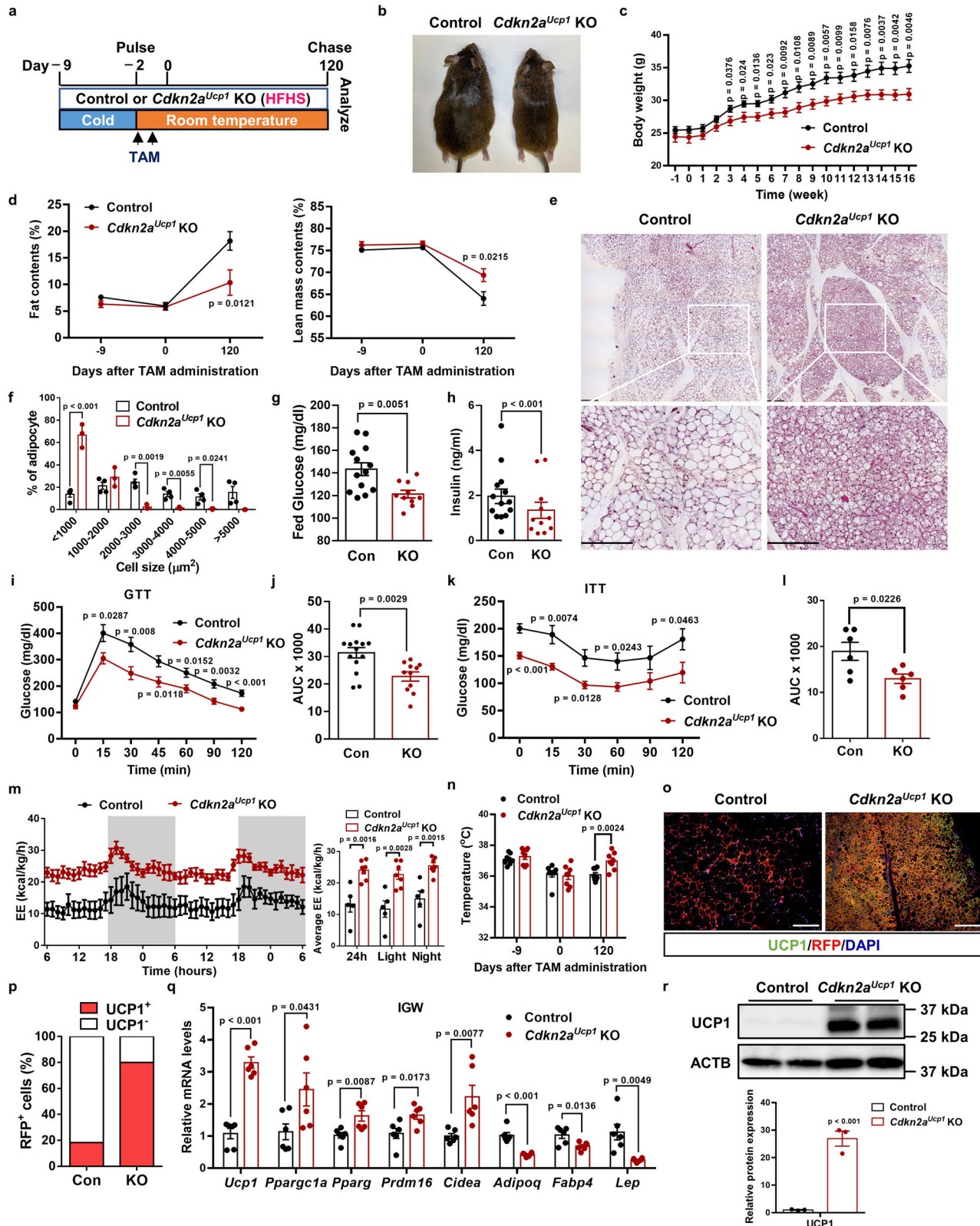

examine directly if the higher number of UCP1-expressing RFP+ cells in *Cdkn2a^{Ucp1}* KO was derived from increased proliferative events, we first assessed the proliferation of control and *Cdkn2a^{Ucp1}* KO cells during beige fat maintenance by bromodeoxyuridine (BrdU) staining. We observed that *Cdkn2a* knockout did not affect beige adipocyte proliferation in our in vitro cellular system (Supplementary Fig. 5a). We

To further rule out the possibility that beige adipocyte maintenance is regulated in a cell cycle-dependent manner, we examined the role of a downstream gene of *Cdkn2a*, *Ccnd1*, which also plays a key role in the regulation of cell cycle. We found that the mRNA level of *Ccnd1* was unchanged during the beige-to-white adipocyte transition (Supplementary Fig. 5b). To further investigate the effect of

**Fig. 2 | Prolonged maintenance of beige fat by deleting *Cdkn2a* prevents HFHS-induced obesity and glucose intolerance by enhancing energy expenditure.**
**a** Experimental procedure. Two-month-old UCP1-RFP (control) and *Cdkn2a^Ucp1* KO male mice were housed under cold conditions for 7 consecutive days to induce beige fat formation. Mice were then administered TAM and fed a HFHS diet at room temperature for 120 days. **b** Representative images of 6-month-old control or *Cdkn2a^Ucp1* KO male mice under HFHS. **c** Body weight of control ($n = 13$) or *Cdkn2a^Ucp1* KO male mice under HFHS ($n = 10$). **d** NMR analysis of fat (left panel) and lean masses (right panel) of body weight of control or *Cdkn2a^Ucp1* KO male mice under HFHS ($n = 7–15$). **e** Tile scan of IGW from control or *Cdkn2a^Ucp1* KO male mice under HFHS. Scale bar, 100 μM. **f** Quantification of adipocyte sizes in IGW from control ($n = 4$) or *Cdkn2a^Ucp1* KO male mice ($n = 3$). **g** The blood glucose level of control ($n = 13$) or *Cdkn2a^Ucp1* KO male mice ($n = 10$) ad libitum feeding with HFHS. **h** The serum insulin level of control ($n = 14$) or *Cdkn2a^Ucp1* KO ($n = 11$) male mice

under HFHS. **i** The blood glucose level of control ($n = 14$) or *Cdkn2a^Ucp1* KO ($n = 11$) male mice under HFHS after intraperitoneal injection of glucose for glucose (GTT) tolerance tests. **j** Area under the curve (AUC) analyses of (**i**). **k** The blood glucose level of control or *Cdkn2a^Ucp1* KO mice under HFHS after intraperitoneal injection of insulin for insulin (ITT) tolerance tests ($n = 6$). **l** Area under the curve (AUC) analyses of (**k**). **m** Energy expenditure of control ($n = 5$) or *Cdkn2a^Ucp1* KO ($n = 7$) male mice fed a HFHS. **n** Body temperature of control or *Cdkn2a^Ucp1* KO male mice during this experiment ($n = 8$). **o** Immunofluorescence staining of UCP1 and RFP in IGW from control or *Cdkn2a^Ucp1* KO male mice. Scale bar, 50 μM. **p** Quantification of the percentage of RFP+ cells that express endogenous UCP1. **q** qPCR analysis of the mRNA expression of thermogenic genes in IGW from control or *Cdkn2a^Ucp1* KO male mice ($n = 6$). **r** Western blot analysis of UCP1 in IGW from control or *Cdkn2a^Ucp1* KO male mice ($n = 3$). *p* values were determined by two-tailed Student's *t* test. Data are expressed as means ± SEM.

*Ccnd1*-mediated cell cycle progression on beige-to-white adipocyte transition in vitro, we constructed an adipocyte-specific conditional *Ccnd1* overexpression (OE) mouse model (Supplementary Fig. 5c, d). After doxycycline (Dox) treatment, we confirmed that the *Ccnd1* expression was significantly higher in *Ccnd1* OE beige adipocytes compared to control cells (Supplementary Fig. 5e). Overexpression of *Ccnd1* did not influence beige adipocyte maintenance, as assessed by immunostaining and western blot analysis of UCP1 (Supplementary Fig. 5f, g). Together, these findings strongly suggest that beige adipocyte maintenance in *Cdkn2a^Ucp1* KO mice is not primarily mediated by the activation of cell proliferation.

### Knockout of *Cdkn2a* promotes beige fat maintenance by inhibiting autophagy

A previous study showed that autophagy-induced mitochondrial clearance (mitophagy) plays a key role in regulating the beige-to-white adipocyte transition[10]. Since *Cdkn2a* has been strongly associated with autophagy in various cell types[23–25], we examined the relationship between *Cdkn2a* and autophagy during beige fat maintenance. Compared to control cells, the number of LC3 puncta, a marker of autophagy, was significantly decreased in *Cdkn2a^Ucp1* KO beige adipocytes after withdrawing external stimuli (Fig. 4a, b). We also found that autophagy was activated in control beige adipocytes during the beige-to-white fat transition, as determined by an increased LC3-II/LC3-I ratio and decreased SQSTM1/p62 (a protein specifically degraded in lysosomes) (Fig. 4c). In contrast, *Cdkn2a* deficiency inhibited autophagy activation at the final stage (Fig. 4c), and this autophagy inhibition was further boosted by treatment with the lysosomal inhibitor chloroquine (CQ) (Fig. 4d). By co-staining with Mitotracker and anti-LC3 antibody, we found that a lower level of mitophagy contributed to the higher mitochondrial content in *Cdkn2a^Ucp1* KO beige adipocytes compared to control cells (Fig. 4e). In agreement with this result, *Cdkn2a* knockout led to an increased mtDNA copy number compared to controls (Fig. 4f). At the molecular level, *Cdkn2a^Ucp1* KO beige adipocytes exhibited higher protein abundance of mitochondrial markers VDAC and TOMM20 (Fig. 4g). In addition, the LC3-II/LC3-I ratio was similar in control and *Ccnd1* OE beige adipocytes after withdrawing external stimuli (Supplementary Fig. 5f), suggesting that *Ccnd1* does not affect autophagy during beige adipocyte maintenance. These results indicate that loss of *Cdkn2a* suppresses autophagy during beige adipocyte maintenance.

To test whether *Cdkn2a* regulated beige adipocyte maintenance in an autophagy-dependent manner, we treated control and *Cdkn2a^Ucp1* KO mice with rapamycin, an autophagy inducer (Fig. 4h). We found that rapamycin treatment accelerated the beige-to-white adipocyte transition in control mice (Fig. 4i), indicating a negative correlation between autophagy activation and beige adipocyte maintenance. Moreover, rapamycin treatment blocked the prolonged beige adipocyte maintenance in IGW from the *Cdkn2a^Ucp1* KO mice (Fig. 4i). Consistent with the in vivo data, rapamycin treatment in vitro also largely hindered *Cdkn2a^Ucp1* KO-mediated beige adipocyte maintenance based

on the number of UCP1-expressing RFP+ cells (Fig. 4j, k). Furthermore, the upregulated expression of UCP1 and other thermogenic genes in *Cdkn2a^Ucp1* KO beige adipocytes was restored to the levels of control cells by rapamycin treatment (Fig. 4l, m). We confirmed that rapamycin treatment induced autophagy as expected in both control and *Cdkn2a^Ucp1* KO cells, as assessed by an increased LC3-II/LC3-I ratio (Fig. 4m). Additionally, rapamycin treatment reversed the enhanced oxygen consumption rate induced by *Cdkn2a* knockout (Fig. 4n). Together, these in vivo and in vitro results suggest that *Cdkn2a* deficiency prevents beige-to-white adipocyte transition by suppressing autophagy.

### *Cdkn2a* promotes autophagy by increasing *Becn1* expression

To investigate how CDKN2A affects autophagy, we tested the effect of *Cdkn2a* knockout on the mRNA levels of autophagy-related genes in vivo and in vitro. We found that *Cdkn2a* deficiency reduced the gene expression of *Becn1* in both IGW and beige adipocytes, while the expression of other autophagy-related genes, including *Atg5*, *Atg12*, *Pink1*, *Bnip3*, *Bnip3l*, and *Fundc1*, was unchanged (Fig. 5a and Supplementary Fig. 6a). The protein expression of BECN1 was increased in the control group after withdrawing external stimuli (Fig. 5b and Supplementary Fig. 6b). Compared to the control group, the *Cdkn2a* knockout group exhibited lower protein levels of BECN1 in vivo and in vitro (Fig. 5b and Supplementary Fig. 6b). Furthermore, we found that the mRNA level of *Becn1* was significantly higher in IGW than in BAT (Supplementary Fig. 6c). Consistently, thermogenic brown and beige adipocytes expressed lower *Becn1* (Supplementary Fig. 6d), indicating a negative correlation between *Becn1* and thermogenesis. Similar to the expression pattern of *p16^Ink4a* and *p19^Arf*, the gene expression of *Becn1* was increased after withdrawing external stimuli (Fig. 5c). Interestingly, the deletion of *Cdkn2a* in brown adipocytes also inhibited *Becn1* expression in vivo and in vitro (Supplementary Fig. 6e, f).

To test whether *Cdkn2a* within beige adipocytes regulated autophagy via *Becn1*, we performed in vitro beige maintenance assays using a previously established hyperactive BECN1 mouse model induced by a single knock-in mutation (F121A) that decreases its interaction with the negative regulator BCL2[26] (Supplementary Fig. 6g). We examined whether *Cdkn2a^Ucp1* KO beige adipocytes could still be maintained in a hyperactive BECN1 setting (Fig. 5d). After a short 20-day rewarming period following cold exposure, we found that hyperactive BECN1 blunted *Cdkn2a^Ucp1* knockout--mediated beige adipocyte maintenance based on H&E analysis (Fig. 5e). We further performed RFP and UCP1 immunostaining on IGW sections and confirmed that upon hyperactive BECN1 (*Cdkn2a^Ucp1* KO + *Becn1*^F121A), *Cdkn2a^Ucp1* KO beige adipocytes started to lose UCP1 expression at day 20 of the rewarming period, with a similar percentage of UCP1-expressing RFP+ cells (Fig. 5f, g) and comparable levels of thermogenic gene expression within IGW (Fig. 5h, i) as the control group. In agreement with these mouse models, we confirmed that the autophagy level was restored to the control level in *Cdkn2a^Ucp1* KO + *Becn1*^F121A mice compared to control

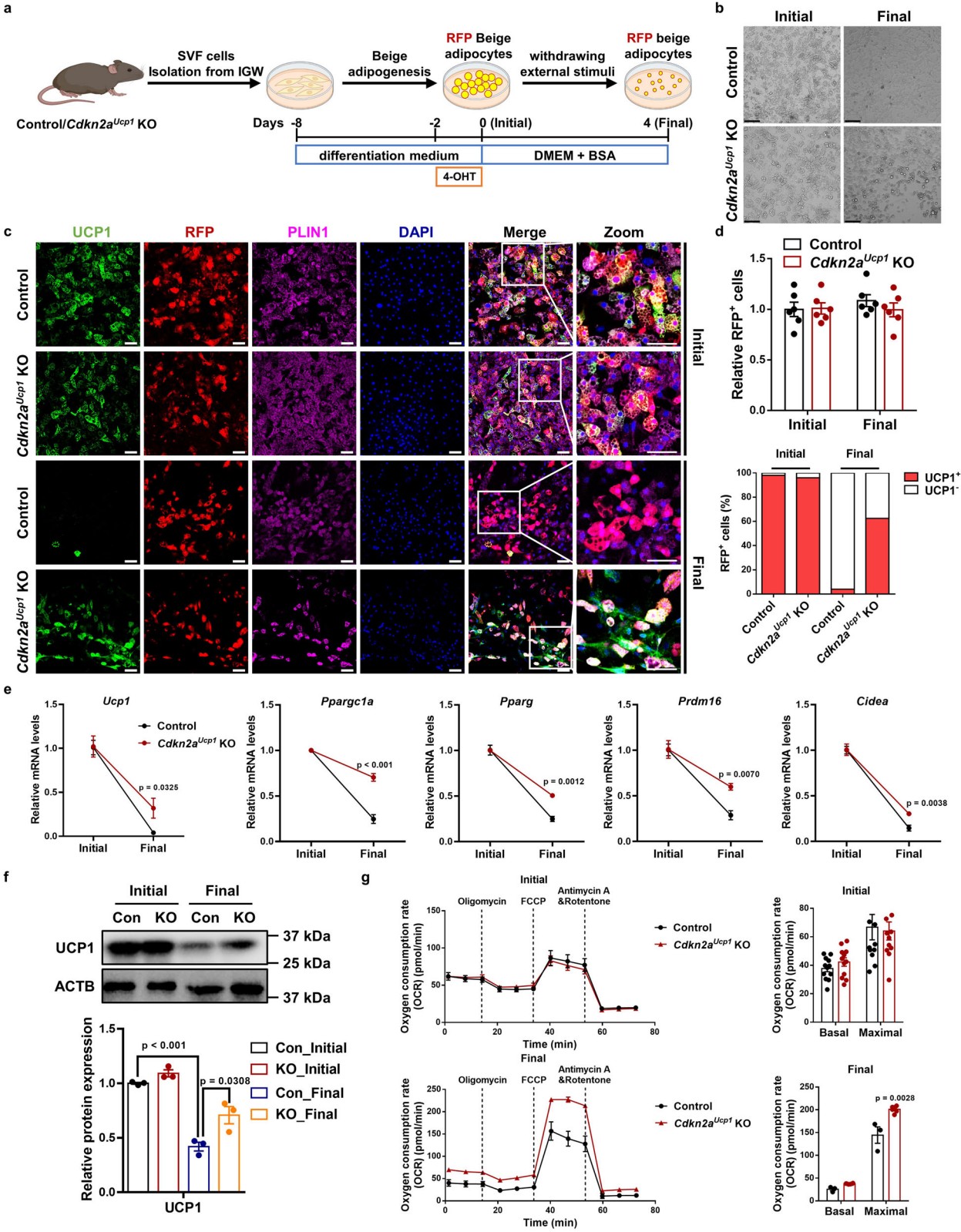

and *Cdkn2a*[Ucp1] KO mice (Fig. 5i), as determined by western blot analysis of the LC3-II/LC3-I ratio. Consistent with our in vivo data, hyperactive BECN1 also impaired *Cdkn2a* knockout-mediated beige adipocyte maintenance in vitro, as assessed by quantification of UCP1-expressing RFP[+] cells (Fig. 5j, k). Additionally, the mRNA expression of UCP1 and other thermogenic genes in the *Cdkn2a*[Ucp1] KO beige adipocytes was restored to the levels in the control mice after hyperactive BECN1

(Fig. 5l, m). Together, these findings suggest that *Cdkn2a* regulates beige adipocyte maintenance through BECN1-mediated autophagy.

### Hyperactive BECN1 accelerates the loss of beige adipocyte characteristics upon removal of external stimuli in vitro

To investigate the role of BECN1 in beige fat maintenance, we isolated SVF cells from *Becn1*[F121A] mice and performed in vitro studies (Fig. 6a).

**Fig. 3 | *Cdkn2a* deficiency promotes beige adipocyte maintenance in vitro.**
**a** Schematic illustration of the cellular system of beige adipocyte maintenance. SVF cells isolated from IGW of control or *Cdkn2a*$^{Ucp1}$ KO male mice were differentiated into beige adipocytes and treated with 4-OHT to induce gene deletion and RFP labeling. Then, external stimuli were withdrawn by changing the differentiation medium to DMEM containing 5% BSA for 4 days. **b** Bright field images of control or *Cdkn2a*$^{Ucp1}$ KO beige adipocytes during beige adipocyte maintenance. Scale bar, 100 μM. **c** Immunofluorescence staining of UCP1 and RFP in control and *Cdkn2a*$^{Ucp1}$ KO beige adipocytes during beige adipocyte maintenance. Scale bar, 50 μM.

**d** Quantification of RFP+ cells (upper panel) and the percentage of RFP+ cells that express endogenous UCP1 (lower panel) (*n* = 6). **e** qPCR analysis of the mRNA expression of thermogenic genes in control and *Cdkn2a*$^{Ucp1}$ KO beige adipocytes during beige adipocyte maintenance. **f** Western blot analysis of UCP1 in control and *Cdkn2a*$^{Ucp1}$ KO beige adipocytes during beige adipocyte maintenance. **g** Seahorse analysis of oxygen consumption rates (OCR) in control and *Cdkn2a*$^{Ucp1}$ KO beige adipocytes during beige adipocyte maintenance (*n* = 12). *p* values were determined by two-tailed Student's *t* test. Data are expressed as means ± SEM of triplicate tests.

We first confirmed that hyperactive BECN1 increased the number of LC3 puncta and the LC3-II/LC3-I ratio, hallmarks of enhanced autophagy, in beige adipocytes compared to control cells (Fig. 6b–d). We found that hyperactive BECN1 did not affect beige adipogenesis (Fig. 6e, f) or UCP1 expression at the initial stage (day 0) (Fig. 6g, h). At the final stage (day 2), hyperactive BECN1 impaired beige adipocyte maintenance and decreased thermogenic gene expression (Fig. 6g–j). We also investigated the function of BECN1 in brown adipocyte maintenance (Supplementary Fig. 7a). Hyperactive BECN1 also enhanced autophagy in brown adipocytes (Supplementary Fig. 7b, c) but did not affect brown fat biogenesis (Supplementary Fig. 7d, e). Similar to *Cdkn2a* knockout in beige adipocytes, no significant difference was observed in maintenance or thermogenic gene expression between control and *Becn1*$^{F121A}$ brown adipocytes at the final stage (Supplementary Fig. 7f–h), suggesting that BECN1-promoted autophagy was not a regulator of brown adipocyte maintenance. Together, these findings suggest that hyperactive BECN1 impairs the maintenance of beige adipocytes but not brown adipocytes.

To further investigate whether BECN1-mediated autophagy regulates human beige adipocyte maintenance, we isolated SVF cells from human subcutaneous WAT and performed in vitro beige maintenance assays. We induced human beige adipogenesis and treated cells with Tat-Becn1, a peptide known to stimulate autophagy through the mobilization of endogenous BECN1[27], after the withdrawal of stimuli. We validated that Tat-Becn1 enhanced autophagy in human beige adipocytes, as assessed by an increased LC3-II/LC3-I ratio (Fig. 6k). As expected, Tat-BENC1 accelerated the decrease of expression of UCP1 and other thermogenic genes in human beige adipocytes after removing stimuli (Fig. 6k, l). These results indicate that BECN1-mediated autophagy plays a conserved role in regulating beige adipocyte maintenance between humans and mice.

## p16$^{INK4a}$ facilitates *Becn1* expression by preventing its mRNA decay
To explore the underlying mechanisms by which *Cdkn2a* regulates BECN1-mediated autophagy in beige adipocytes, we first examined whether there were any changes in the cellular distributions of p16$^{INK4a}$ and p19$^{ARF}$ during the beige-to-white transition. Intriguingly, the cytoplasmic accumulation of p16$^{INK4a}$ in beige adipocytes was significantly increased 6 h post withdrawing external stimuli, while the cellular distribution of p19$^{ARF}$ was unaltered (Fig. 7a), suggesting that p16$^{INK4a}$ and p19$^{ARF}$ might play distinct roles in mediating *Becn1*. Since a previous mRNA-bound proteome study identified p16$^{INK4a}$ and p19$^{ARF}$ as potential RNA-binding proteins[28], we decided to investigate whether p16$^{INK4a}$ and/or p19$^{ARF}$ regulate *Becn1* expression by binding to its mRNA. We performed RNA immunoprecipitation (RIP)-qPCR with antibodies against endogenous p16$^{INK4a}$ or p19$^{ARF}$ (Fig. 7b). We found that p16$^{INK4a}$ interacted with the mRNA of *Becn1*, but not *Atg5*, *Atg7*, *Pink1*, or *Ucp1*, in beige adipocytes upon withdrawal of external stimuli (Fig. 7c). In addition, no interaction was observed between p19$^{ARF}$ and *Becn1* mRNA (Fig. 7d). Together, these results indicate that p16$^{INK4a}$ regulates *Becn1* by specifically targeting its transcript.

Since p16$^{INK4a}$ positively modulated *Becn1* gene expression and p16$^{INK4a}$ was involved in the regulation of mRNA stability[29], we investigated the *Becn1* mRNA stability in control and *Cdkn2a*$^{Ucp1}$ KO beige

adipocytes by using an RNA decay assay. As expected, *Cdkn2a* deficiency markedly decreased the mRNA stability of *Becn1*, but not *Ucp1*, in beige adipocytes (Fig. 7e). Collectively, these results demonstrate that p16$^{INK4a}$ serves as an RNA-binding protein during the beige-to-white transition and regulates *Becn1* expression by mediating its mRNA stability.

## p19$^{ARF}$ promotes BECN1-mediated autophagy by interacting with BCL2L1
A previous study reported that p19$^{ARF}$ reduced the formation of the protein complex between BECN1 and its negative regulator, BCL2L1, and then induced autophagy in cancer cells[23]. Here, we asked whether p19$^{ARF}$ might interfere with the ability of BCL2L1 to complex with BECN1 protein, leading to enhanced autophagy during the beige-to-white transition. To test this hypothesis, we first detected the interaction among p19$^{ARF}$, BECN1 and BCL2L1 in beige adipocytes. Using a co-IP assay, we found that p19$^{ARF}$ could interact with BECN1 and BCL2L1 (Fig. 7f). In contrast, p16$^{INK4a}$ was unable to interact with BECN1 or BCL2L1 (Fig. 7g). In the reciprocal IP, p19$^{ARF}$ was also pulled down by an antibody against BCL2L1 (Fig. 7h). Importantly, the complex formation between BECN1 and BCL2L1 was markedly increased in *Cdkn2a*$^{Ucp1}$ KO beige adipocytes compared to control cells (Fig. 7h). Taken together, these results indicate that p19$^{ARF}$ enhances BECN1-mediated autophagy by interacting with BCL2L1.

## *Cdkn2a* and *Becn1* expression are positively associated with obesity in mice and humans
We further investigated the correlation between *Cdkn2a* and *Becn1* mRNA levels and obesity. Upon HFHS feeding, the mRNA levels of *p16*$^{Ink4a}$, *p19*$^{Arf}$, and *Becn1* in IGW were significantly higher than in mice fed a standard chow diet (Fig. 8a). Most remarkably, the *p16*$^{INK4a}$, *p14*$^{ARF}$, and *BECN1* mRNA levels in subcutaneous adipose tissues of obese human individuals were significantly higher than that of non-obese individuals (Fig. 8b), linking *p16*$^{INK4a}$, *p14*$^{ARF}$, and *BECN1* expression to obesity in human. Furthermore, we found that higher mRNA levels of *p16*$^{INK4a}$ and *p14*$^{ARF}$ were associated with higher body weight, BMI, and fat mass in humans (Fig. 8c, d). *BECN1* expression was positively correlated with body weight, BMI, fat mass, and blood glucose levels (Fig. 8e). Thus, these data indicate that *CDKN2A and BECN1* mRNA levels could be indicators of human obesity and T2DM.

## Discussion
Since the rediscovery of brown and beige adipocytes in adults, tremendous efforts have been made to find signals to induce beige adipocyte formation within WAT. In this study, we used unique mouse models to demonstrate that beige adipocytes can be formed and maintained by pharmacological and genetic methods. The main finding of this study is that *Cdkn2a* is a crucial negative regulator of beige adipocyte maintenance through direct interaction with *Becn1*. We report that genetically deleting *Cdkn2a* in beige adipocytes promotes their lifespan and is associated with an improved metabolic profile upon HFHS diet challenge. Our data reveal a previously unknown function of *Cdkn2a* in beige fat biology and suggest that inhibitors of the *Cdkn2a* pathway would be beneficial in reversing the deleterious effects of metabolic syndromes. Therefore, it serves as a potential

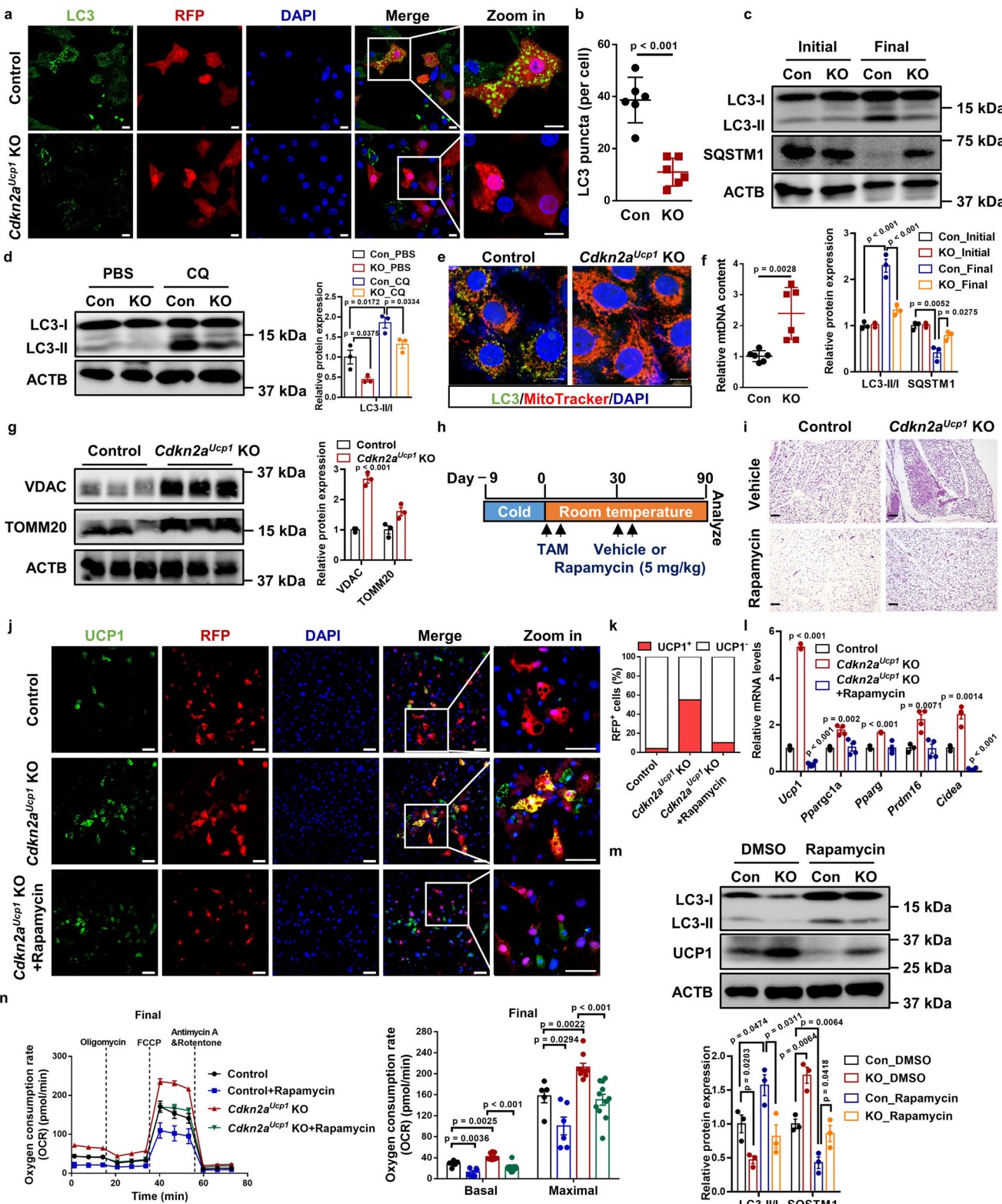

therapeutic target in treating obesity and its related metabolic diseases.

Activation of beige fat biogenesis is accompanied by improved metabolic benefits. There is, however, little evidence that maintaining beige adipocyte phenotype can be beneficial. In this study, we investigated whether sustained beige adipocytes can improve metabolic fitness under diet-induced obesity, thereby achieving anti-obesity and anti-diabetes effects. We found that *Cdkn2a* deletion promotes beige

adipocyte maintenance under both a standard chow and a HFHS diet. Importantly, the retained beige adipocytes, induced by UCP1[+]-cell-specific *Cdkn2a* deletion, are adequate to prevent body weight gain, fat expansion, and ectopic lipid storage in the liver under HFHS feeding. Along with the presence of preserved beige adipocytes, the mice also have improved glucose tolerance. These results are intriguing, given the recent clinical observation that a strong association exists between greater brown fat activity and lower cardiovascular risk, such as high

**Fig. 4 | *Cdkn2a* deletion promotes beige fat maintenance by inhibiting autophagy. a** Immunofluorescence staining of LC3 and RFP in control and *Cdkn2a*[Ucp1] KO beige adipocytes after withdrawing external stimuli. Scale bar, 10 μM. **b** Quantification of the number of LC3 puncta per cell (*n* = 6). **c** Western blot analysis of LC3, SQSTM1 in control and *Cdkn2a*[Ucp1] KO beige adipocytes during beige adipocyte maintenance. **d** Western blotting analysis of LC3 in control and *Cdkn2a*[Ucp1] KO beige adipocytes after withdrawing external stimuli and treated with or without 10 μM chloroquine (CQ). **e** Immunofluorescence staining of MitoTracker and LC3 in control and *Cdkn2a*[Ucp1] KO beige adipocytes after withdrawing external stimuli. Scale bar, 10 μM. **f** The relative mtDNA copy number in control and *Cdkn2a*[Ucp1] KO beige adipocytes after withdrawing external stimuli. **g** Western blot analysis of VDAC and TOMM20 in control and *Cdkn2a*[Ucp1] KO beige adipocytes after withdrawing external stimuli. **h** Experimental procedure. Two-month-old control and *Cdkn2a*[Ucp1] KO male mice were housed under cold conditions for 7 consecutive days to induce beige fat formation and then administered vehicle or 5 mg/kg rapamycin at room temperature for 90 days. **i** Representative H&E staining images of IGW from control or *Cdkn2a*[Ucp1] KO male mice treated with or without rapamycin. Scale bar, 100 μM. **j** Immunofluorescence staining of UCP1 and RFP in control and *Cdkn2a*[Ucp1] KO beige adipocytes treated with or without 10 nM rapamycin after withdrawing external stimuli. Scale bar, 50 μM. **k** Quantification of the percentage of RFP+ cells that express endogenous UCP1. **l** qPCR analysis of the mRNA expression of thermogenic genes in control and *Cdkn2a*[Ucp1] KO beige adipocytes treated with or without 10 nM rapamycin after withdrawing external stimuli (*n* = 3–4). **m** Western blot analysis of UCP1 and LC3 in control and *Cdkn2a*[Ucp1] KO beige adipocytes treated with or without 10 nM rapamycin after withdrawing external stimuli. **n** Seahorse analysis of oxygen consumption rates (OCR) in control and *Cdkn2a*[Ucp1] KO beige adipocytes treated with or without 10 nM rapamycin after withdrawing external stimuli (*n* = 9–11). *p* values were determined by two-tailed Student's *t* test. Data are expressed as means ± SEM of triplicate tests.

blood pressure and coronary artery disease[30]. Several recent studies have shown that beige and brown fat can influence systemic metabolism through releasing endocrine signals, including peptides, lipids, miRNAs, and secreted proteins, independent of non-shivering thermogenesis[31,32]. Our data suggest that signals released from *Cdkn2a*[Ucp1] KO beige adipocytes may communicate with other organs to maintain whole-body energy metabolism.

One critical obstacle in beige adipocyte research is that there are no specific Cre lines to precisely target brown vs. beige adipocytes in mouse models. Deletion of *Cdkn2a* by UCP1-Cre[ERT2] inevitably creates a model with gene knockout in brown fat, which may confound the phenotype. However, in our *Cdkn2a*[Ucp1] KO model, we did not observe BAT phenotypes based on morphology and thermogenic gene expression profile, indicating that the resulting metabolic benefits may largely come from beige adipocyte maintenance. Interestingly, although our data suggest that BAT maintenance is independent of *Cdkn2a* and *Becn1*, the regulation of *Becn1* expression by *Cdkn2a* is essentially conserved in brown and beige adipocytes both in vivo and in vitro. It is thus conceivable that, in addition to *Cdkn2a* and *Becn1*, multiple complex mechanisms may exist to retain BAT identity and their physiological roles. Combined with the prior knowledge that beige adipocytes are developmentally, molecularly, and functionally different from brown adipocytes[33], our finding substantiates this important concept that brown and beige adipocyte identities are different and are also distinctly regulated.

Mechanistically, we found that *Cdkn2a* deficiency mediates beige adipocyte maintenance by regulating autophagy and is independent of *Ccnd1* activity and likely independent of cell cycle control. During the beige-to-white transition, autophagy genes have been shown to contribute to beige adipocyte "whitening"[10,11,34–36]. Interestingly, we found that rapamycin further decreased *Ucp1* and *Cidea* expression compared to the control mice. As rapamycin is an inhibitor of mTORC1, which is essential for promoting beige adipogenesis[37,38], it is possible that rapamycin repressed beige adipocyte activity by inhibiting the mTORC1 pathway. Consistently, early studies using adipose-specific deletion of autophagy-related genes show that mice accumulate more mitochondria in WAT, producing a browning of WAT phenotype with reduced body weight and WAT mass[39,40]. Among the autophagy pathway, BECN1 is a scaffolding protein that is key for the initiation of autophagosome biogenesis[41], and therefore, we can target and activate autophagy at early steps; the role of BECN1 in adipose tissue has been recently appreciated: adipocyte-specific BECN1 knockout mice develop severe lipodystrophy and metabolic dysfunctions[42,43]. However, it is not clear what role BECN1 plays in UCP1+ cells and whether BECN1 mediates beige activity downstream of the *Cdkn2a* pathway. In this study, we identified the Cdkn2a pathway as an upstream mechanism that cooperates with autophagy regulation to govern beige-to-white adipocyte transition. More specifically, p16[INK4a] facilitates *Becn1* expression by its function in preventing its mRNA decay, and p19[ARF] promotes BECN1-mediated autophagy by interacting with its negative regulator BCL2L1.

Thus, our findings provide mechanistic insights into key players in beige fat maintenance regulation.

Therapeutically, the expressions of both *Cdkn2a* and *Becn1* are upregulated in obesity and positively associated with adiposity in mice and humans. Our findings reveal that *Cdkn2a* and *Becn1* function as potent positive regulators of beige-to-white adipocyte transition, providing a potential therapeutic target for pharmacological intervention to combat the obesity epidemic and its related metabolic diseases. Since direct manipulation of either *Cdkn2a* or *Becn1* could cause several downstream effects and/or potential tumor formation, we believe that defining specific upstream and downstream factors of the *Cdkn2a*-*Becn1* pathway will pave the way to develop more focused treatments for obesity and metabolic diseases. Previously, we and others have demonstrated that the ability to form beige adipocytes declines when mice grow older[44–51]. Therefore, it will be interesting to test whether manipulating *Cdkn2a* or *Becn1* still maintains their beiging capacity and whether beige adipocytes can be maintained in the older stage. The possibility that the beiging potential can be retained in obese and aged humans and animals suggests a potential therapeutic approach to use beige adipocytes against obesity in older adults.

In summary, we have identified *Cdkn2a* as an important player in beige adipocyte maintenance via Becn1-mediated autophagy. Our work creates a concept that beige adipocytes can stay long-lived after discontinuing the stimuli and could be developed as an alternative therapeutic target for treating obesity by improved energy expenditure and metabolic fitness.

## Methods

### Mouse models

Mice were housed at 22 ± 2 °C with a humidity of 35 ± 5% and a 14:10 light:dark cycle with a standard rodent chow diet and water unless otherwise indicated. *Rosa26R*[RFP] (Stock No. 007914) mice were obtained from the Jackson Laboratory. *Ucp1*-Cre[ERT2] and *Cdkn2a*[fl/fl] mice were generously provided by Dr. Eric N. Olson (University of Texas Southwestern Medical Center). *Becn1*[F121A] mice were generously provided by Dr. Congcong He (Northwestern University). All mice were maintained on mixed C57BL6/J-129SV background. All animal experiments were performed according to procedures reviewed and approved by the Institutional Animal Care and Use Committee of the University of Illinois at Chicago. Mice were euthanized by carbon dioxide asphyxiation ($CO_2$) inhalation and cervical dislocation was performed as a secondary euthanasia procedure.

### Generation of TRE-Ccnd1 mice

Mice were generated as previously described[52]. Briefly, we introduced p2Lox-*Ccnd1* into ZX1 ES cells and selected for G418-resistant recombinant colonies, in which Ccnd1 was recombined downstream of a tetracycline-responsive promoter (TRE). ZX1 ES cell clones were then validated in culture, and three clones were used to generate three

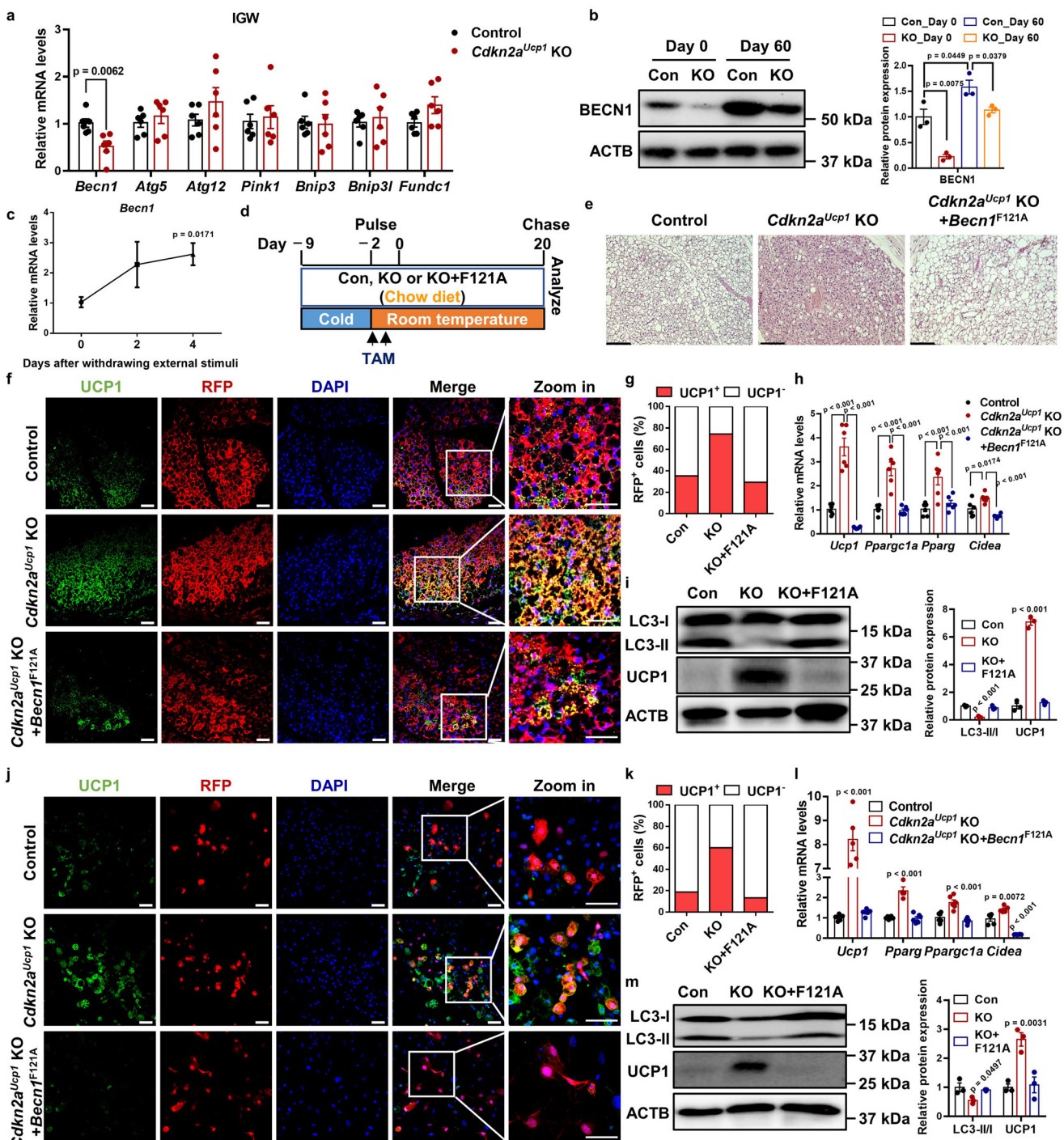

**Fig. 5 | *Cdkn2a* regulates autophagy by positively mediating *Becn1* expression in vivo and in vitro. a** qPCR analysis of the mRNA expression of autophagy-related genes in IGW from control or *Cdkn2a*$^{Ucp1}$ KO male mice fed a HFHS (*n* = 6).
**b** Western blot analysis of BECN1 in IGW from control or *Cdkn2a*$^{Ucp1}$ KO male mice. **c** Expression profile of *Becn1* during beige adipocyte maintenance. **d** Experimental procedure. Two-month-old control, *Cdkn2a*$^{Ucp1}$ KO and *Cdkn2a*$^{Ucp1}$ KO + *Becn1*$^{F121A}$ male mice were housed under cold conditions for 7 consecutive days to induce beige fat formation. Mice were then administered tamoxifen (TAM) and fed a chow diet at room temperature for 20 days. **e** Representative H&E staining images of IGW from control, *Cdkn2a*$^{Ucp1}$ KO or *Cdkn2a*$^{Ucp1}$ KO + *Becn1*$^{F121A}$ male mice after cold stimulus withdrawal. Scale bar, 100 μM. **f** Immunofluorescence staining of UCP1 in IGW from control, *Cdkn2a*$^{Ucp1}$ KO or *Cdkn2a*$^{Ucp1}$ KO + *Becn1*$^{F121A}$ male mice after cold stimulus withdrawal. Scale bar, 50 μM. **g** Quantification of the percentage of RFP+ cells that express endogenous UCP1. **h** qPCR analysis of the mRNA expression of

thermogenic genes in IGW from control, *Cdkn2a*$^{Ucp1}$ KO or *Cdkn2a*$^{Ucp1}$ KO + *Becn1*$^{F121A}$ male mice after cold stimulus withdrawal (*n* = 6). **i** Western blot analysis of UCP1 and LC3 in IGW from control, *Cdkn2a*$^{Ucp1}$ KO or *Cdkn2a*$^{Ucp1}$ KO + *Becn1*$^{F121A}$ male mice after cold stimulus withdrawal. **j** Immunofluorescence staining of UCP1 and RFP in control and *Cdkn2a*$^{Ucp1}$ KO beige adipocytes with or without hyperactive BECN1 after withdrawing external stimuli. Scale bar, 50 μM. **k** Quantification of the percentage of RFP+ cells that express endogenous UCP1. **l** qPCR analysis of the mRNA expression of thermogenic genes in control and *Cdkn2a*$^{Ucp1}$ KO beige adipocytes with or without hyperactive BECN1 after withdrawal of external stimuli (*n* = 4–6). **m** Western blot analysis of UCP1 and LC3 in control and *Cdkn2a*$^{Ucp1}$ KO beige adipocytes with or without hyperactive BECN1 after withdrawing external stimuli. *p* values were determined by two-tailed Student's *t* test. Data are expressed as means ± SEM of triplicate tests.

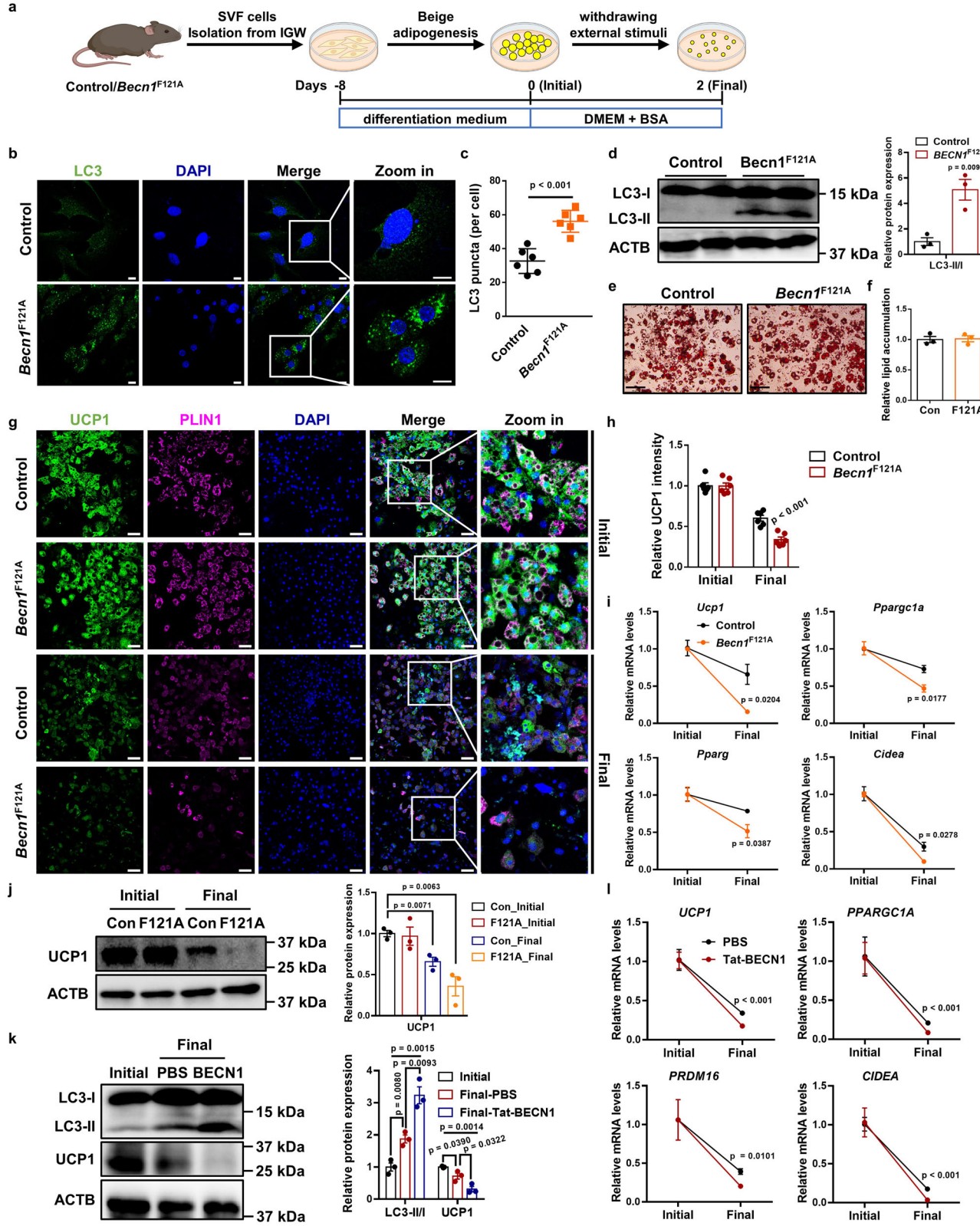

chimeric mouse lines. All three-mouse strains (TRE-*Ccnd1*) behaved similarly and had similar induction of *Ccnd1* expression in response to doxycycline.

## Cell culture
The isolated cells were cultured in DMEM/F12 media (Sigma-Aldrich) supplemented with 10% FBS (Sigma-Aldrich) and 1% Penicillin/

Streptomycin (Gibco) at 37 °C in a 5% $CO_2$ humidified incubator. For beige and brown adipocyte differentiation, post-confluent cells were induced by DMEM/F12 containing 10% FBS, 0.5 mM iso-butylmethylxanthine (Sigma-Aldrich), 1 μg/ml insulin (Sigma-Aldrich), 1 μM dexamethasone (Sigma-Aldrich), 60 μM indomethacin (Sigma-Aldrich), 1 nM triiodo-L-thyronine (Sigma-Aldrich), and 1 μM rosiglita-zone (Sigma-Aldrich) for 2 days and with DMEM/F12 containing 10%

**Fig. 6 | BECN1 does not influence beige biogenesis but accelerates the loss of beige adipocyte characteristics after withdrawal of external stimuli in vitro. a** Schematic illustration of the cellular system of beige adipocyte maintenance. SVF cells isolated from IGW of control or *Becn1*[F121A] male mice were differentiated into beige adipocytes. Then, external stimuli were withdrawn by changing the differentiation medium to DMEM containing 5% BSA for 2 days. **b** Immunofluorescence staining of LC3 in control and *Becn1*[F121A] beige adipocytes. Scale bar, 10 μM. **c** Quantification of the number of LC3 puncta per cell (*n* = 6). **d** Western blot analysis of LC3 in control and *Becn1*[F121A] beige adipocytes. **e** Oil red O staining of control and *Becn1*[F121A] beige adipocytes before withdrawing external stimuli. Scale bar, 100 μM. **f** Relative lipid accumulation was quantified with a microplate spectrophotometer. **g** Immunofluorescence staining of UCP1 and PLIN1 in control and *Becn1*[F121A] beige adipocytes during beige adipocyte maintenance. Scale bar, 50 μM. **h** Quantification of the percentage of UCP1-expressing cells (*n* = 6). **i** qPCR analysis of the mRNA expression of thermogenic genes in control and *Becn1*[F121A] beige adipocytes during beige adipocyte maintenance. **j** Western blot analysis of UCP1 in control and *Becn1*[F121A] beige adipocytes during beige adipocyte maintenance. **k** Western blot analysis of UCP1 and LC3 in human beige adipocytes treated with or without 5 μM Tat-BECN1 after withdrawal of external stimuli. **l** qPCR analysis of the mRNA expression of thermogenic genes in human beige adipocytes treated with or without 5 μM Tat-BECN1 after withdrawal of external stimuli. *p* values were determined by two-tailed Student's *t* test. Data are expressed as means ± SEM of triplicate tests.

FBS and 1 μg/ml insulin, 1 nM triiodo-L-thyronine, and 1 μM rosiglitazone every 2 days. Fresh medium was replaced every 2 days until ready for harvest. To withdraw external stimuli, the differentiation medium was changed to DMEM containing 5% BSA for 4 days.

### Tamoxifen treatment
To induce Cre recombination, mice were intraperitoneally injected with 1.5 mg/kg body weight of tamoxifen (TAM; Cayman Chemical) dissolved in sunflower oil (Sigma-Aldrich) for 2 consecutive days. For Cre induction in vitro, cells were treated with 2 μM 4-hydroxy-Tamoxifen (4-OHT, Sigma-Aldrich).

### Cold exposure experiment
For cold exposure experiments, mice were placed in a 6 °C environmental chamber (Environmental & Temperature Solutions) for 7 days. Body temperature was measured using a rectal probe (Physitemp). The probe was lubricated with glycerol and inserted 1.27 centimeters (0.5 inch), and the temperature was recorded when stabilized at the indicated time points.

### Metabolic cage studies
Mice were housed individually and acclimatized to the metabolic chambers (Promethion System, Sable System International) at the UIC Biologic Resources Laboratory for 2 days before data collection was initiated. For the subsequent 3 days, food intake, VO$_2$, VCO$_2$, energy expenditure and physical activity were monitored over a 12 h light/dark cycle with food provided ad libitum. For energy expenditure analysis, embedded ANCOVA tools in a web application CalR[53] were used to perform regression-based indirect calorimetric analysis. For body composition analysis, the total fat and lean mass were assessed with Bruker Minispec 10 whole body composition analyzer (Bruker).

### Glucose and insulin tolerance tests
For the glucose tolerance test, mice fasted for 16 h were intraperitoneally injected with glucose (1.5 g/kg). For insulin tolerance test, mice fasted for 4 h were intraperitoneally injected with human insulin (1.0 U/kg). Glucose levels were measured in tail blood at 0, 15, 30, 45, 60, 90 and 120 min after glucose injection by using a glucose meter (Contour Next).

### Isolation of stromal vascular fraction (SVF)
Isolation of SVF cells from IGW and BAT was performed as previously described[20]. Briefly, IGW or BAT was excised from 6-week-old male mice, minced by scissors and then digested in isolation buffer (100 mM HEPES, 0.12 M NaCl, 50 mM KCl, 5 mM D-glucose, 1.5% BSA, 1 mM CaCl$_2$, pH 7.3) containing 1 mg/ml type I collagenase (Worthington) at 37 °C for 30 min. Digested tissue was filtered through 70 μm mesh to remove large pieces, and the flow-through was then centrifuged at 800 × *g* for 5 min. Pellets containing primary preadipocytes were resuspended in red blood cell lysis buffer (155 mM NH$_4$Cl, 10 mM KHCO$_3$, 0.1 mM EDTA) for 5 min, followed by centrifugation at 800 × *g* for 5 min. The pellets were resuspended

and seeded. Human SVF cells were isolated from fresh subcutaneous adipose tissues.

### Western blot analysis
Proteins from cells and tissue were extracted with RIPA buffer (Boston BioProducts) supplemented with protease inhibitor cocktail (MedChem Express) and centrifuged at 12,000 × *g* for 15 min at 4 °C. The total protein amounts were detected by a BCA assay kit (Thermo Fisher Scientific). Protein samples were separated in an SDS-PAGE gel and transferred to PVDF membranes (Millipore). The membranes were blocked with 5% non-fat milk at room temperature and then incubated with relevant primary antibodies at 4 °C overnight, followed by HRP-conjugated secondary antibody at room temperature for 2 h. The signals were detected by the addition of Western ECL Substrate (Thermo Fisher Scientific). The protein bands were detected using ChemiDoc MP Image system (Biorad). The primary antibodies used in this study are listed in Supplementary Table 1.

### Histological staining
Hematoxylin and eosin (H&E) staining was performed on paraffin sections using standard methods. Briefly, tissues were fixed in 10% formalin overnight, dehydrated, embedded in paraffin, and sectioned with a microtome at 4–8 μm thicknesses. Adipocyte sizes were quantified by using ImageJ. For immunohistochemistry (IHC), sections were deparaffinized, boiled in antigen-retrieval solution, treated with primary antibodies in blocking buffer (5% donkey serum) at 4 °C overnight, treated with secondary antibody for 2 h at room temperature in blocking buffer, and stained with Vectastain ABC KIT (Vector Laboratories) and DAB KIT (Vector Laboratories). Immunostaining images were taken using a Leica DMi8 microscope (Leica). For immunofluorescence staining, paraffin sections were incubated with permeabilization buffer (0.3% Triton X-100 in PBS) for 30 min at room temperature, with primary antibody at 4 °C overnight, and with secondary antibody for 2 h at room temperature, all in blocking buffer (5% donkey serum in 1X PBS). Images were taken using a confocal laser microscope (Carl Zeiss Ltd).

### Oil red O staining
Cells were washed and fixed in 4% paraformaldehyde at room temperature for 1 h. After washing with 60% isopropanol three times, the cells were stained with 60% filtered Oil Red O working solution (vol/vol in distilled water) of Oil red O stock solution (Sigma-Aldrich) at room temperature for 15 min. Cells were washed with ddH$_2$O before imaging. To quantify lipid accumulation, Oil Red O-stained lipids were eluted in 100% isopropanol, and then the optical density (OD) was measured at 500 nm.

### RNA isolation and quantitative real-time PCR (qPCR) analysis
Total RNA from tissue or cells was isolated using Tripure Isolation Reagent (Sigma-Aldrich) and Bullet Blender Homogenizer (Next Advance) according to the manufacturer's protocol. cDNA was converted from 2 μg of total RNA by using a High Capacity cDNA Reverse

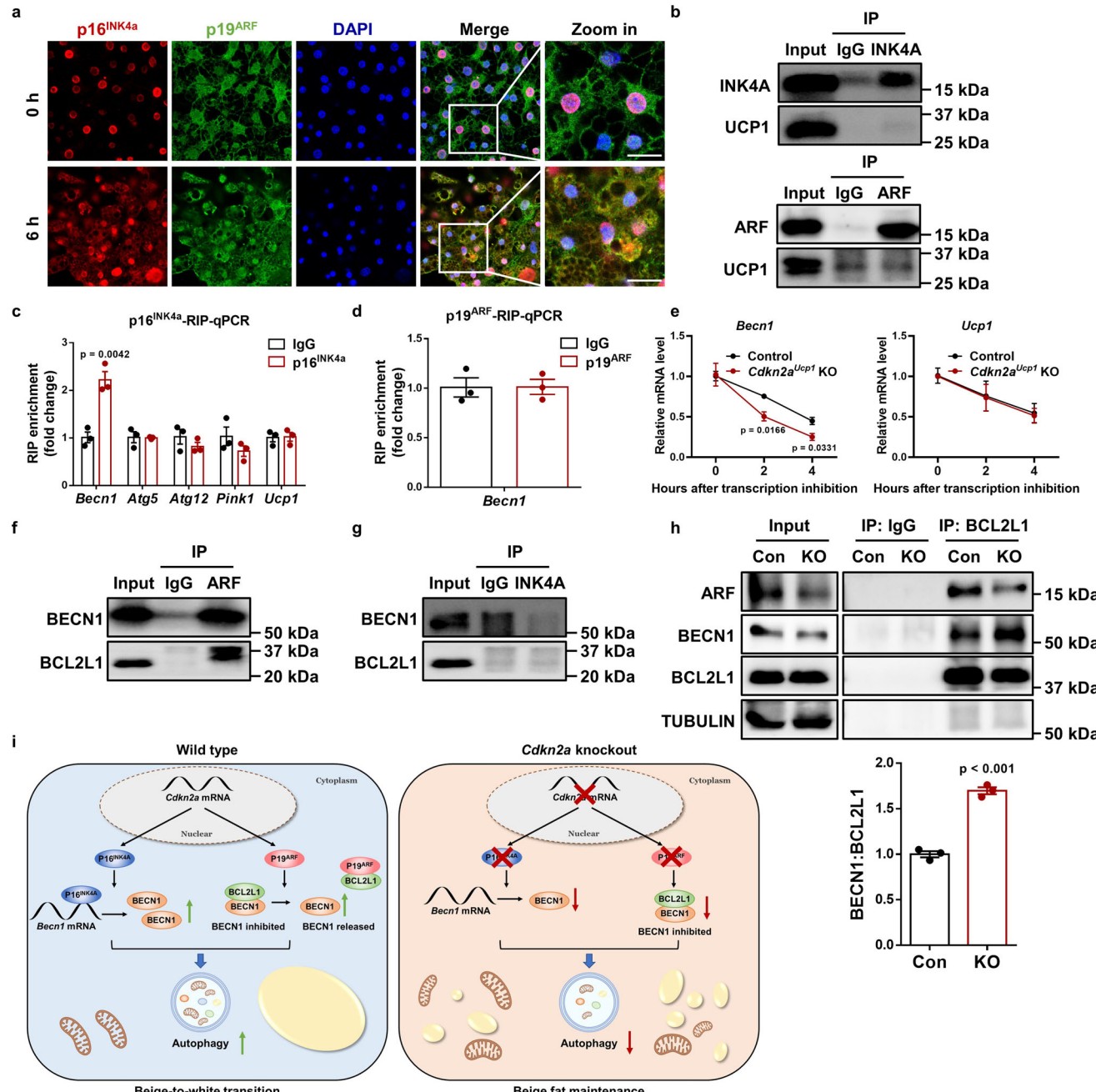

**Fig. 7 | Cdkn2a positively regulates Becn1 expression by suppressing its mRNA decay and promotes BECN1-mediated autophagy by interacting with BCL2L1.** **a** Immunofluorescence staining of p16^INK4a and p19^ARF distribution during beige adipocyte maintenance. **b** Western blot to confirm the immunoprecipitation of p16^INK4a and p19^ARF proteins. **c** RNA immunoprecipitation-qPCR (RIP-qPCR) analysis of the interaction of *Becn1*, *Atg5*, *Atg12*, *Pink1* and *Ucp1* transcripts with p16^INK4a in beige adipocytes (n = 3). **d** RNA immunoprecipitation-qPCR (RIP-qPCR) analysis of the interaction of the *Becn1* transcript with p19^ARF in beige adipocytes (n = 3). **e** mRNA lifetimes of *Becn1* and *Ucp1* in control and *Cdkn2a^Ucp1* KO beige adipocytes (n = 3). **f** Co-immunoprecipitation (Co-IP) analysis of the interaction between p19^ARF and BECN1 or BCL2L1. **g** Co-IP analysis of the interaction between p16^INK4a and

BECN1 or BCL2L1. **h** Co-IP analysis of the interaction between BCL2L1 and BECN1 in control and *Cdkn2a^Ucp1* KO beige adipocytes after withdrawing external stimuli (n = 3). Quantification of the BECN1/BCL2L1 ratio in the IP samples is shown in the right panel. **i** Working model. In wild type cells, p16^INK4a specifically targets *Becn1* mRNA and increases its expression by promoting mRNA stability, while p19^ARF interacts with BCL2L1 and releases BECN1, thereby enhancing autophagy and beige-to-white transition. Loss of *Cdkn2a* decreases the protein levels of p16^INK4a and p19^ARF, resulting in reduced expression and activity of BECN1, thereby inhibiting autophagy and promoting beige fat maintenance. *p* values were determined by two-tailed Student's *t* test. Data are expressed as means ± SEM.

Transcription Kit (Thermo Fisher Scientific). qPCR was performed using 2X Universal SYBR Green Fast qPCR Mix (Abclonal) following the manufacturer's instructions and analyzed with a ViiA7 system (Applied Biosystems). Data were analyzed using the comparative Ct method and the relative expression of mRNAs was determined after normalization to *RplpO*. Primer sequences are available in Supplementary Table 2.

## Oxygen consumption rate (OCR) analysis
The oxygen consumption rate of beige adipocytes was measured by using a Seahorse XF96 Extracellular Flux Analyzer according to the manufacturer's instructions. SVF cells were seeded on XF96 and induced to differentiate and maintain. The plate was subjected to a temperature-controlled (37 °C) extracellular analyzer at indicated time

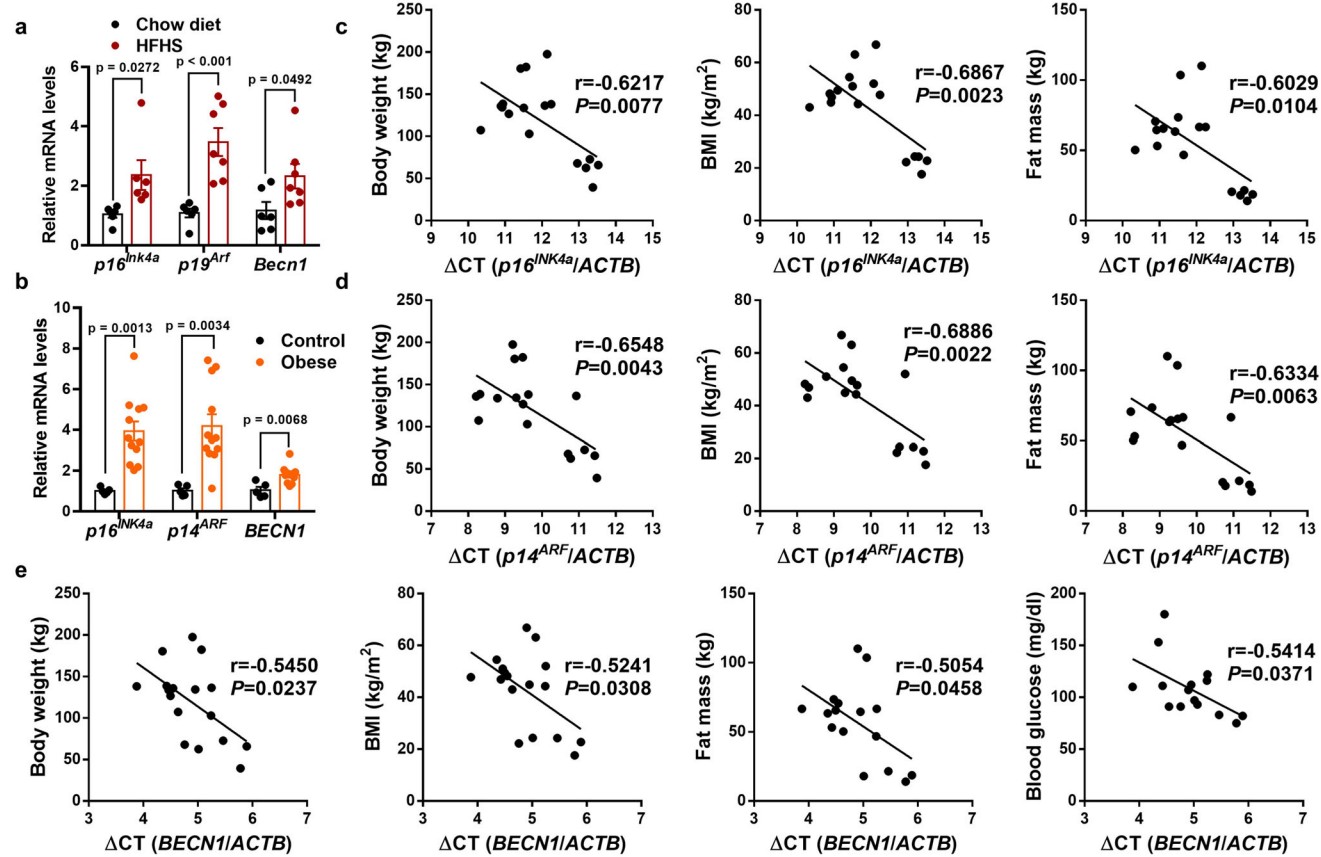

**Fig. 8 | *Cdkn2a* and *Becn1* expression levels positively correlate with obesity in mice and humans. a** qPCR analysis of the mRNA expression of *p16^Ink4a^*, *p19^Arf^* and *Becn1* in IGW from male mice fed on chow diet or HFHS (*n* = 6–7). **b** qPCR analysis of the mRNA expression of *p16^Ink4a^*, *p14^Arf^* and *Becn1* in subcutaneous WAT from obese and non-obese people (*n* = 5–12). **c** Two-tailed Pearson's correlation analysis of *p16^INK4a^* expression in human subcutaneous WAT and body weight, BMI or fat mass

of 17 individuals. **d** Two-tailed Pearson's correlation analysis of *p14^ARF^* expression in human subcutaneous WAT and body weight, BMI or fat mass of 17 individuals. **e** Two-tailed Pearson's correlation analysis of *BECN1* expression in human subcutaneous WAT and body weight, BMI, fat mass and blood glucose levels of 15–17 individuals. *p* values were determined by two-tailed Student's *t* test. Data are expressed as means ± SEM.

point. After measuring initial oxygen consumption rate (OCR), 4 μM oligomycin, 2 μM FCCP and 1 μM rotenone/actinomycin A were then sequentially added into the plate by automatic pneumatic injection. Data were analyzed by Seahorse Wave software. Basal respiration was calculated as [OCR^initial^ – OCR^R&A^]. The maximum respiration rate was computed as [OCR^FCCP^ – OCR^R&A^].

### Measurement of mitochondrial DNA (mtDNA)

The total DNA was isolated from beige adipocytes by using DNA isolation kit. mtDNA was amplified using primers specific for the mitochondrial cytochrome c oxidase subunit 2 (Cox2) gene and normalized to an intron of the nuclear-encoded Hbb (β-globin) gene as described before[54]. The primer sequences were listed in Supplementary Table 2.

### RNA immunoprecipitation-qPCR (RIP-qPCR)

RIP was performed by using a Dynabeads Protein A immunoprecipitation kit (Thermo Fisher Scientific) according to the manufacturer's instructions. Briefly, mature adipocytes were lysed in lysis buffer (150 mM KCl, 10 mM HEPES, 2 mM EDTA, 0.5% NP-40, 0.5 mM DTT, protease inhibitor cocktail and RNase inhibitor) for 30 min at 4 °C. The lysates were centrifuged, and the supernatant was collected. A small aliquot of lysate was saved as input, and the remaining sample was incubated with Protein A magnetic beads conjugated with p16^INK4a^, p19^ARF^ or IgG antibody at 4 °C for 4 h. Subsequently, the beads were washed with wash buffer and eluted in elution buffer. The input and immunoprecipitated RNAs were isolated by Tripure Isolation Reagent

and reverse transcribed into cDNA. The fold enrichment was detected by qPCR.

### Co-immunoprecipitation (Co-IP)

Co-IP was performed by using a Pierce crosslink magnetic IP/Co-IP kit (Thermo Fisher Scientific) according to the manufacturer's instructions. Cells were rinsed with cold PBS, lysed in IP lysis buffer, cleared by centrifugation and protein concentration was estimated. A small aliquot of lysate was saved as input, and the remaining sample was incubated with Protein A magnetic beads conjugated with p16^INK4a^, p19^ARF^, BCL2L1 or IgG antibody at 4 °C overnight. Then, the beads were washed with wash buffer and eluted by heating in loading buffer. Immunoprecipitates and input were electrophoresed, transferred to PVDF membranes, and probed with the indicated antibodies.

### RNA stability assay

RNA stability assay was performed as previously described[55]. Briefly, cells were treated with 5 μg/ml actinomycin D (Sigma-Aldrich) to inhibit global mRNA transcription. Samples were collected at the indicated time points. Total RNA was extracted for reverse transcription, and the levels of genes of interest were analyzed by qPCR.

### Human participants

Study participants were recruited from the Bariatric and General Surgery Clinics at the University of Illinois Hospital. We enrolled twelve obese subjects (8 females and 4 males) who underwent laparoscopic gastric sleeve bariatric surgery and five lean healthy controls (3 females

and 2 males) who underwent elective surgeries such as hernia repair and abdominal wall reconstruction. Subcutaneous adipose tissue samples were obtained during surgery. Adipose tissue samples were snap-frozen and stored for biological analyses. The age of the participants ranged from 21 to 47 years old. Exclusion criteria included chronic inflammatory diseases, chronic organ failure, autoimmune diseases, cancer of any type, current smoking, or current pregnancy. Subjects who were deemed eligible were informed about the study details, risks, and precautions taken to reduce this risk. All the study participants provided informed written consent. Physical characteristics, including body weight and body mass index (BMI), were measured. Total fat mass was assessed using dual x-ray absorptiometry (DXA; iDXA, General Electric Inc). Plasma concentrations of glucose were quantified in the fasting state as we previously described[56]. All protocols and procedures of the study followed the standards set by the latest version of the Declaration of Helsinki and were approved by the Institutional Review Board of The University of Illinois at Chicago (protocol code 2017-1125). The details of population characteristics are listed in Supplementary Table 3.

### Statistics and reproducibility
All data are presented as mean ± SEM. Two-tailed unpaired Student's *t* test (for comparison of two groups) or one-way ANOVA followed by Tukey's test (for comparison of three or more groups) were conducted using GraphPad Prism software. All experiments were repeated two or three times with representative data or images shown. $p < 0.05$ was considered statistically significant in all the experiments.

### Reporting summary
Further information on research design is available in the Nature Portfolio Reporting Summary linked to this article.

## Data availability
The data that support the findings of this study are available in the "Methods" and Supplementary Material of this article. Additional data are available upon request from the corresponding author. Source data are provided with this paper.

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

## Acknowledgements

We thank Dr. Lijun Rong for help with manuscript preparation and giving fruitful advice. We thank members in the Y.J. laboratory, especially Nipun Velupally and Meram Mohamed Mahmoud, for mouse genotyping and technical help. We thank Cynthia Rose Adams and Jeanette Purcell for assistance with mouse husbandry, Stefan J. Green and the Research Resources Center for RT-PCR analysis, Dr. Brian Layden and Metabolic Phenotyping Core for analytical and phenotypical mouse measurements, Balaji Ganesh and Flow Cytometry Core facility for FACS and members of the Y.J. laboratory for helpful comments on the manuscript. This work was supported by grants to Y.J. from the National Institute of Diabetes and Digestive and Kidney Disease grant (NIDDK) K01 DK11177, R03 DK127149, R01 DK132398 and Pilot & Feasibility Diabetes Research & Training Center (DRTC) Award (P30DK020595). Cartoons in Figs. 3a, 6a, 7i and Supplementary Figs. 4a, 5d, 7a were created with BioRender.com.

## Author contributions

Y.J. conceived and designed the experiments. R.W. and J.P. conducted most of the experiments. R.W., J.P., and Y.J. interpreted the experiments. R.W.A. and M.K. developed the TRE-Ccnd1 transgenic mouse line. A.M.M. provided the human adipose samples. R.W. and Y.J. wrote and revised the manuscript, and J.P., Y.Q., Z.S., R.H., Y.Y., S.X., Z.W., G.Y., S.G.O., Q.S., Z.S., A.M.M., P.X., C.H., R.W.A., M.K., G.S., and Q.J. reviewed it.

## Competing interests

The authors declare no competing interests.
