## [Peer Review File · Nature Communications]

Genetically prolonged beige fat in male mice confers long-lasting metabolic healthREVIEWER COMMENTS

Reviewer #1 (Remarks to the Author):

This study addresses a key barrier to beige adipocyte translation – maintenance as beige adipocytes are quickly inactivated after stimuli removal. There are numerous studies on the activation/origin of beige adipocytes, but the maintenance is much less understood. The authors employed a series of delicate novel mouse models and developed an in vitro beige adipocyte maintenance model to demonstrate that Cdkn2a deficiency is critical to maintain beige adipocytes after cold exposure and therefore induces attractive metabolic improvements in body weight, energy expenditure, and glucose tolerance. They further elucidated a mechanism through suppressing BECN1-mediated autophagy by combining mouse genetic, pharmaceutical, and molecular approaches. Moreover, the translational implications of their findings were demonstrated in human clinical data and primary adipocyte culture. Overall, this is an exciting and novel finding to advance beige adipocyte study into a new phase. The experiments are well designed and carried out in high rigor, with a very fluent presentation of the data.

Comments:

1. Beige fat activation needs sympathetic neural input. In the Cdkn2a KO mice with prolonged beige fat maintenance, is there increased neural signal or innervation to sustain their higher activity?
2. The description of no morphological difference in the BAT of Cdkn2a KO mice is not accurate. They showed less lipid accumulation on chow feeding (Ext data Fig. 1b) and on HFD feeding (Ext data Fig. 2e).
The largely blunted phenotype is carefully discussed. However, there seem lower knockout efficiency in BAT than in IGW (comparing Ext data Fig. 1f vs. Ext data Fig. 2i). Is it due to different diets? What's the knockout efficiency in BAT on chow then?
3. The energy expenditure is not different between light and dark phases. Please double check.
4. Are blood glucose levels in Fig. 2g at fasting or ad libitum? Please include serum insulin levels to strengthen the improved insulin sensitivity in the obese Cdkn2a KO mice.
5. In Fig. 5i, rapamycin further repressed Ucp1 and Cidea expression than the control mice (rather than normalize). It suggests other mechanism of rapamycin than through Cdkn2a in repressing beige adipocyte activity. Please at least discuss this point.
6. The prolonged beige fat maintenance in the Cdkn2a KO mice is observed after beige fat activation. It therefore suggests that only inhibiting Cdkn2a pathway is not sufficient to introduce the metabolic benefits and combined treatment with browning reagent might be needed.
7. In Fig. 5a, please include mouse condition (chow?) in the legend.
8. Please include mouse genetic background information.

Reviewer #2 (Remarks to the Author):

The manuscript entitled "Genetically prolonged beige fat in mice confers long-lasting metabolic health" by Riufan Wu et al., describes the role of the tumor suppressor gene Cdkn2a in beige adipocytes maintenance. The authors identify a novel mechanism in which the Cdkn2a products, p16 and p19, modulate autophagy through the control of RNA decay and protein binding of Becn1 gene and Bcl2L1 protein, respectively. This leads to decreases autophagy in Cdkn2a-UCP1 KO mice and maintenance of the beige phenotype, protecting mice from diet induced obesity. In addition, the authors correlate BECN1, p16 and p14 to increased BMI, fat mass and blood glucose in humans. The experiments are well controlled and demonstrate, through the use of genetically relevant mouse

models and pharmacological approaches, the hypothesis raised by the authors. These are important novel data for the obesity field that may help to design new potential therapeutic strategies, although caution should be taken when manipulating the expression of a tumor suppressor gene (which is discussed in the manuscript).

There are some points that need clarification.

Have the authors used male mice? What is the phenotype in female? What is the genetic background of the mice?

The use of rapamycin is interesting. However, this is not highly specific. Do the authors try more specific small molecules?

Have the authors measured insulinemia and/or insulin sensitivity in their mouse models?

Reviewer #3 (Remarks to the Author):

The manuscript presented by Wu et al. investigates the mechanisms through which *cdkn2a* drives the transition from beige to white adipocytes after cold exposure. In general terms, the study tests a novel and exciting idea. Also, the manuscript is well-written, organized, and the figures look well.

I have some comments that could help you to support your conclusions better.

1. Having experience on UCP1 western blot in IGW, It seems surprising to me the strong bands you get even in some control conditions. I have experienced some unspecific (strong) bands with some UCP1 antibodies. Could you please, provide some positive and negative control? This comment is across several figures.

2. Even the most convincing evidence of changes in autophagy comes from in-vivo autophagy flux (leupeptin); the in-vitro autophagy flux (+/- autophagy inhibitor) could be enough to claim the direction of autophagy only when several replicates are tested and ideally (in the context of this paper) by immunofluorescence (LC3 puncta) and western blot. You only show one WB, no replicates, in fig 4d.

3. Across several figures, there are representative WBs without quantification. Do you have replicates? Please provide at least 3 replicates, plus quantification and statistical analysis.

4. You mention in the text that the role of autophagy in the transition from beige to white adipocytes is driven by the degradation of mitochondria (mitophagy). Considering that,

a) Could you measure some markers of mitophagy? This data will support the degradation of mitochondria by autophagy.

b) In the same line, it will be exciting to see if markers of mitochondrial mass are consistent with this idea.

5. The molecular weight (kDa) should be present in every western blot image to discriminate the band's position.

Point-by-point response

We appreciate the Reviewers' recognition of the significance and rigor of this study. We also thank the Reviewers for their thoughtful comments and helpful points greatly improving our manuscript.

Reviewer #1 (Remarks to the Author):

This study addresses a key barrier to beige adipocyte translation – maintenance as beige adipocytes are quickly inactivated after stimuli removal. There are numerous studies on the activation/origin of beige adipocytes, but the maintenance is much less understood. The authors employed a series of delicate novel mouse models and developed an in vitro beige adipocyte maintenance model to demonstrate that *Cdkn2a* deficiency is critical to maintain beige adipocytes after cold exposure and therefore induces attractive metabolic improvements in body weight, energy expenditure, and glucose tolerance. They further elucidated a mechanism through suppressing BECN1-mediated autophagy by combining mouse genetic, pharmaceutical, and molecular approaches. Moreover, the translational implications of their findings were demonstrated in human clinical data and primary adipocyte culture. Overall, this is an exciting and novel finding to advance beige adipocyte study into a new phase. The experiments are well designed and carried out in high rigor, with a very fluent presentation of the data.

Comments:

1. Beige fat activation needs sympathetic neural input. In the *Cdkn2a* KO mice with prolonged beige fat maintenance, is there increased neural signal or innervation to sustain their higher activity?

Response: We agree with the Reviewer that neural innervation could contribute to beige fat maintenance. Therefore, we conducted additional experiments to investigate whether increased sympathetic neural signaling was responsible for the prolonged beige fat maintenance in *Cdkn2a^{Ucp1}* KO mice. To test this hypothesis, we performed western blot analysis against tyrosine hydroxylase (TH), which is a marker of norepinephrine turnover. However, our results did not reveal any significant difference in TH expression between the control and *Cdkn2a^{Ucp1}* KO mice in the IGW at Day 60 (please refer to Fig. 1).

2. 1) The description of no morphological difference in the BAT of *Cdkn2a* KO mice is not accurate. They showed less lipid accumulation on chow feeding (Ext data Fig. 1b) and on HFD feeding (Ext data Fig. 2e).

Response:

1) We apologize for the confusion. After double-checking our data, we can confirm that there was no significant morphological difference in BAT between control and *Cdkn2a*^{Ucp1} KO mice fed on chow diet (see Ext data Fig. 1d). To avoid any potential confusion, we have replaced the images in Ext data Fig. 1d. However, under HFHS condition, we found that *Cdkn2a*^{Ucp1} KO did indeed ameliorate HFHS-induced BAT whitening and lipid accumulation (see Ext data Fig. 2e).

2) The largely blunted phenotype is carefully discussed. However, there seem lower knockout efficiency in BAT than in IGW (comparing Ext data Fig. 1f vs. Ext data Fig. 2i). Is it due to different diets? What's the knockout efficiency in BAT on chow then?

2) We apologize for the confusion. Our qPCR analysis indicated that the knockout efficiency of *Cdkn2a* was very similar in both BAT (Ext Data Fig. 1h) and IGW (Fig. 1i), with a 90% reduction in expression levels. We want to clarify that the data presented in both Ext Data Fig. 1h (previously labeled as Ext Data Fig. 1f) and Ext Data Fig. 2i were analyzed using both BAT. In Ext Data Fig. 1h, we confirmed that the expression of *Cdkn2a* (both p16Ink4a and p19Arf) was significantly reduced in BAT from *Cdkn2a*^{Ucp1} KO mice at Day 0 under a standard chow diet. In Ext Data Fig. 2i, we observed that the reduction of p16Ink4a and p19Arf in

BAT remained but was less extensive after the mice were fed a high-fat, high-sugar (HFHS) diet for 120 days at room temperature. We think this might be due to complementary mechanisms or expression from non-UCP1 cells after the prolonged feeding of the HFHS diet for 120 days. Indeed, our own data have shown that the expression of *Cdkn2a* can be induced by a longer HFD treatment.

Ext data Fig. 1h (previously labeled as Ext Data Fig. 1f), qPCR analysis of mRNA expression of thermogenic genes in BAT from control or *Cdkn2a^{Ucp1}* KO male mice at day 0 (left panel) and day 60 (right panel) post withdrawal cold stimulus.

Ext data Fig. 2i, qPCR analysis of mRNA expression of thermogenic genes in BAT from control or *Cdkn2a^{Ucp1}* KO male mice under HFHS at day 120 post withdrawal cold stimulus.

3. The energy expenditure is not different between light and dark phases. Please double check.
 Response: Thank you for your comments. We have carefully re-examined our data on energy expenditure and re-analyzed it. We found that the energy expenditure during the dark phase was significantly higher than that during the light phase. The results are shown in Fig 2m. (see below)

Fig. 2m, Energy expenditure of control or *Cdkn2a^{Ucp1}* KO male mice fed a HFHS.

4. Are blood glucose levels in Fig. 2g at fasting or ad libitum? Please include serum insulin levels to strengthen the improved insulin sensitivity in the obese Cdkn2a KO mice.

Response: 1) We appreciate your comments. To clarify, we measured glucose levels in mice that were fed ad libitum with HFHS diet. To avoid any potential confusion, we have added the phrase "ad libitum feeding with HFHS" to the legend of Fig. 2g.

2) We have measured serum insulin level in $Cdkn2a^{Ucp1}$ KO mice and found that KO mice exhibited a significant decrease in serum insulin levels compared to control mice under HFHS conditions (Fig. 2h). Moreover, the loss of Cdkn2a improved insulin sensitivity compared to control mice, as assessed by the insulin tolerance test (Fig. 2k, l). These findings suggest that $Cdkn2a^{Ucp1}$ KO mice exhibit improved insulin sensitivity compared to control mice.

Fig. 2h, The serum insulin level of control or $Cdkn2a^{Ucp1}$ KO male mice under HFHS.

Fig. 2k, The blood glucose level of control or $Cdkn2a^{Ucp1}$ KO mice under HFHS after intraperitoneal injection of insulin for insulin (ITT) tolerance tests.

Fig. 2l, Area under the curve (AUC) analyses of (k).

5. In Fig. 5i, rapamycin further repressed Ucp1 and Cidea expression than the control mice (rather than normalize). It suggests other mechanism of rapamycin than through Cdkn2a in repressing beige adipocyte activity. Please at least discuss this point.

Response: We thank the Reviewer for pointing this out. We agree with the reviewer and we have revised the Discussion section of our manuscript to include the following statement.

“Interestingly, we found that rapamycin further decreased Ucp1 and Cidea expression compared to the control mice. As rapamycin is an inhibitor of mTORC1, which is essential for promoting beige adipogenesis^{37,38}, it is possible that rapamycin repressed beige adipocyte activity by inhibiting the mTORC1 pathway”. Reference:

37 Liu, D. X. et al. Activation of mTORC1 is essential for beta-adrenergic stimulation of adipose browning. *Journal Of Clinical Investigation* 126, 1704-1716, doi:10.1172/Jci83532 (2016).

38 Liu, D. X., Ceddia, R. P. & Collins, S. Cardiac natriuretic peptides promote adipose 'browning' through mTOR complex-1. *Molecular Metabolism* 9, 192-198, doi:10.1016/j.molmet.2017.12.017 (2018).

6. The prolonged beige fat maintenance in the Cdkn2a KO mice is observed after beige fat activation. It therefore suggests that only inhibiting Cdkn2a pathway is not sufficient to introduce the metabolic benefits and combined treatment with browning reagent might be needed.

Response: The Reviewer raises an interesting question about studying the metabolic differences between control and Cdkn2a^{Ucp1} KO mice without the use of browning reagents. However, it is important to note that the number of pre-existing UCP1+ beige adipocytes within inguinal white adipose tissue is limited without the use of browning reagents. Therefore, we expect that any metabolic differences observed without the use of a browning reagent will not be significant. As a result, all the mice in our study were first exposed to cold (browning reagent) for a week to induce the formation of a large amount of beige fat, which was then used to study its maintenance and metabolic differences.

7. In Fig. 5a, please include mouse condition (chow?) in the legend.

Response: In Fig. 5a, we measured the mRNA expression of autophagy-related genes in IGW from control or KO mice fed a HFHS. The mouse condition has been included in the legend of Fig. 5a.

8. Please include mouse genetic background information.

Response: We have included "All mice were maintained on mixed C57BL6/J-129SV background" to Methods section.

Reviewer #2 (Remarks to the Author):

The manuscript entitled "Genetically prolonged beige fat in mice confers long-lasting metabolic health" by Riufan Wu et al., describes the role of the tumor suppressor gene Cdkn2a in beige adipocytes maintenance. The authors identify a novel mechanism in which the Cdkn2a products, p16 and p19, modulate autophagy through the control of RNA decay and protein binding of Becn1 gene and Bcl2L1 protein, respectively. This

leads to decreases autophagy in *Cdkn2a*-UCP1 KO mice and maintenance of the beige phenotype, protecting mice from diet induced obesity. In addition, the authors correlate BECN1, p16 and p14 to increased BMI, fat mass and blood glucose in humans.

The experiments are well controlled and demonstrate, through the use of genetically relevant mouse models and pharmacological approaches, the hypothesis raised by the authors. These are important novel data for the obesity field that may help to design new potential therapeutic strategies, although caution should be taken when manipulating the expression of a tumor suppressor gene (which is discussed in the manuscript).

There are some points that need clarification.

1) Have the authors used male mice? 2) What is the phenotype in female? 3) What is the genetic background of the mice?

Response:

- 1) Male mice were used in this study.
- 2) Similar to the observations in male mice, female *Cdkn2a*Ucp1 KO mice also exhibited increased beige adipocyte content (Ext Data Fig. 1b) and higher expression levels of thermogenic genes at day 60 compared to control mice (Ext Data Fig. 1c), indicating that the knockout of *Cdkn2a* promotes beige adipocyte maintenance in both male and female mice. These data are now included in the revised manuscript.

3) We have added “All mice were maintained on mixed C57BL6/J-129SV background” to the Methods section in our revised manuscript.

The use of rapamycin is interesting. However, this is not highly specific. Do the authors try more specific small molecules?

Response: Thank you for your comment. We have not yet explored other small molecules beyond rapamycin, as the available specific autophagy activators are currently limited. Therefore, we have shifted our focus to our genetic mouse model, which uses BECN1^{F121A} mice to specifically activate autophagy. This approach allows us to better investigate the role of autophagy in our experimental system.

Have the authors measured insulinemia and/or insulin sensitivity in their mouse models?

Response: We agree and have added the data. Please see above Reviewer 1 point 4. Please see Fig. 2h, 2k, and 2l.

Reviewer #3 (Remarks to the Author):

The manuscript presented by Wu et al. investigates the mechanisms through which cdkn2a drives the transition from beige to white adipocytes after cold exposure. In general terms, the study tests a novel and exciting idea. Also, the manuscript is well-written, organized, and the figures look well. I have some comments that could help you to support your conclusions better.

We thank the Reviewer for their helpful comments and strategies. We believe implementing these strategies improved our study and strengthened our conclusions.

1. Having experience on UCP1 western blot in IGW, It seems surprising to me the strong bands you get even in some control conditions. I have experienced some unspecific (strong) bands with some UCP1 antibodies. Could you please, provide some positive and negative control? This comment is across several figures.

Response:

1) The UCP1 antibody (Thermo Fisher Scientific, PA1-24894) was used in this study.

2) The reason why we saw strong UCP1 bands in IGW across several figures at Day 0 is because all the mice were first cold-pretreated for a week (from Day -9 to Day -2).

3) As per reviewer's suggestion, we have conducted additional experiments to further verify the specificity of the UCP1 antibody. We measured the expression of UCP1 in inguinal white adipose tissue (IGW) from two-month-old male mice at room temperature and after 7 days of cold exposure. Brown adipose tissue (BAT) and perigonadal white adipose tissue (PGW) from mice at room temperature were used as positive and negative controls, respectively. We found that cold exposure significantly increased UCP1 expression in IGW. As expected, we observed strong UCP1 expression in BAT, while UCP1 expression was undetectable in PGW. The results of our experiments are shown below:

2. Even the most convincing evidence of changes in autophagy comes from in-vivo autophagy flux (leupeptin); the in-vitro autophagy flux (+/- autophagy inhibitor) could be enough to claim the direction of autophagy only when several replicates are tested and ideally (in the context of this paper) by immunofluorescence (LC3 punta) and western blot. You only show one WB, no replicates, in fig 4d.

Response: We did WB with 3 replicates and provided quantification and statistical analysis. The results are as follows.

Fig. 4d

3. Across several figures, there are representative WBs without quantification. Do you have replicates? Please provide at least 3 replicates, plus quantification and statistical analysis.

Response: To conserve space in the figures, we only showed the representative bands of WBs. We performed WB with 3 replicates and provided quantification and statistical analysis for each. Quantifications for each WB have been added to the respective figures. The results are as follows.

Fig. 1k

Fig. 1l

Fig. 2r

Fig. 3f

Fig. 4c

Fig. 4d

Fig. 4g

Fig. 4m

Fig. 5b

Fig. 5i

Fig. 5m

Fig. 6d

Fig. 6j

Fig. 6k

Ext data Fig. 1g

Ext data Fig. 5g

Ext data Fig. 6b

4. You mention in the test that the role of autophagy in the transition from beige to white adipocytes is driven by the degradation of mitochondria (mitophagy). Considering that,

a) Could you measure some markers of mitophagy? This data will support the degradation of mitochondria by autophagy. (Tomm20, VADC)

b) In the same line, it will be exciting to see if markers of mitochondrial mass are consistent with this idea.

Response: We thank the Reviewer for this advice. To assess the inhibition of mitophagy in mutant samples, we have measured the expression of mitophagy marker genes, including Pink1, Bnip3, Bnip3l and Fundc1. No significant difference was observed in the expression of mitophagy marker genes between control and *Cdkn2a^{Ucp1}* KO IGW (Fig. 5a) or beige adipocytes (Ext Data Fig. 6a). However, by co-staining with Mitotracker and anti-LC3 antibody, we found that a lower level of mitophagy contributed to a higher mitochondrial content in *Cdkn2a^{Ucp1}* KO beige adipocytes compared to control cells (Fig. 4e, f). Consistently, *Cdkn2a^{Ucp1}* KO beige adipocytes exhibited higher protein abundance of mitochondrial markers VDAC and TOMM20 (Fig. 4g). Collectively, our data suggest that *Cdkn2a^{Ucp1}* KO promotes beige maintenance by inhibiting autophagy-mediated mitochondrial degradation.

Fig. 5a, qPCR analysis of the mRNA expression of autophagy-related genes in IGW from control or *Cdkn2a^{Ucp1}* KO male mice fed a HFHS.

Extended Data Fig. 6a, qPCR analysis of the mRNA expression of autophagy-related genes in control and *Cdkn2a^{Ucp1} KO* beige adipocytes after withdrawing external stimuli.

Fig. 4e, Immunofluorescence staining of MitoTracker and LC3 in control and *Cdkn2a^{Ucp1} KO* beige adipocytes after withdrawing external stimuli. Scale bar, 10 μ M.

Fig. 4f, The relative mtDNA copy number in control and *Cdkn2a^{Ucp1} KO* beige adipocytes after withdrawing external stimuli.

Fig. 4g, Western blot analysis of VDAC and TOMM20 in control and *Cdkn2a^{Ucp1} KO* beige adipocytes after withdrawing external stimuli.

5. The molecular weight (kDa) should be present in every western blot image to discriminate the band's position.

Response: We agree with the Reviewer and have updated the western blot images by including the molecular weight (kDa) markers, which are now displayed in each image.

REVIEWERS' COMMENTS

Reviewer #1 (Remarks to the Author):

The reviewer's comments have been fully addressed.

Reviewer #2 (Remarks to the Author):

The authors have replied to all my concerns, and, based on the new version of the manuscript, the study has been largely improved, including new data that further strengthen the quality of the study.

Reviewer #3 (Remarks to the Author):

The manuscript presented by Wu et al. shows how the deletion of *cdkn2a* in beige adipocytes prolongs the beige phenotype after the stimuli (cold exposure) is removed. In general terms, the study tests a novel and exciting idea. Also, the manuscript is well-written, organized, and the figures look well.

The findings represent an advance in adipose tissue browning as a therapeutic alternative for obesity and related metabolic diseases. The methods are adequate for answering the presented hypothesis, the analysis of the results seems well done, and the conclusions are well supported. The authors addressed all my comments.